# Inductive Gradient Adjustment
# for Spectral Bias in Implicit Neural Representations

**Kexuan Shi** [1]   **Hai Chen** [2][3]   **Leheng Zhang** [1]   **Shuhang Gu** [1]

## Abstract

Implicit Neural Representations (INRs), as a versatile representation paradigm, have achieved success in various computer vision tasks. Due to the spectral bias of the vanilla multi-layer perceptrons (MLPs), existing methods focus on designing MLPs with sophisticated architectures or repurposing training techniques for highly accurate INRs. In this paper, we delve into the linear dynamics model of MLPs and theoretically identify the empirical Neural Tangent Kernel (eNTK) matrix as a reliable link between spectral bias and training dynamics. Based on this insight, we propose a practical **I**nductive **G**radient **A**djustment (**IGA**) method, which could purposefully improve the spectral bias via inductive generalization of eNTK-based gradient transformation matrix. Theoretical and empirical analyses validate impacts of IGA on spectral bias. Further, we evaluate our method on different INRs tasks with various INR architectures and compare to existing training techniques. The superior and consistent improvements clearly validate the advantage of our IGA. Armed with our gradient adjustment method, better INRs with more enhanced texture details and sharpened edges can be learned from data by tailored impacts on spectral bias. The codes are available at: https://github.com/LabShuHangGU/IGA-INR.

## 1. Introduction

The main idea of implicit neural representations (INRs) is using neural networks such as multi-layer perceptrons (MLPs) to parameterize discrete signals in an implicit and continuous manner. Benefiting from the continuity and implicit nature, INRs have gained great attention as a versatile signal representation paradigm (Luo et al., 2025) and achieved state-of-the-art performance across various computer vision tasks such as signal representation (Sitzmann et al., 2020; Liang et al., 2022; Saragadam et al., 2023; Kazerouni et al., 2024; Zhang et al., 2025), 3D shape reconstruction (Zhu et al., 2024; Zhang et al., 2024), and novel view synthesis (Mildenhall et al., 2021; Müller et al., 2022).

However, obtaining high-precision INRs is non-trivial. MLPs with ReLU activations often fail to represent high-frequency details. Such tendency of MLPs to represent simple patterns of target function is referred to as spectral bias (Xu, 2018; Rahaman et al., 2019). To improve the performance of INRs, great efforts have been made to alleviate or overcome this bias of MLPs. These efforts have primarily focused on how MLPs are constructed, such as complex input embeddings (Tancik et al., 2020; Müller et al., 2022; Xie et al., 2023) and sophisticated activation functions (Sitzmann et al., 2020; Ramasinghe & Lucey, 2022; Saragadam et al., 2023; Zhu et al., 2024; Shenouda et al., 2024). Recently, two studies find that training dynamics adjustments based on traditional training techniques, i.e., Fourier reparameterized training (FR) (Shi et al., 2024a) and batch normalization (BN) (Cai et al., 2024), can overcome the spectral bias. Compared to the above approaches emphasizing architectural refinements, these methods could offer considerable representation performance gains while barely increasing the complexity of inference structures.

Despite these works serendipitously revealing the great potential of training dynamics in alleviating spectral bias for better INRs, the underlying mechanism driving their improvements remains unclear. Moreover, there is no clear guidance on the choice of Fourier bases matrix for reparameterization or batch normalization layer to confront spectral bias with varying degrees in various INR tasks. Lack of reliable insight and guidance results in unstable and suboptimal improvements across different scenarios and INR models (Essakine et al., 2024), hindering their broader applications.

Notably, a stream of works (Ronen et al., 2019; Tancik et al., 2020; Shi et al., 2024a; Cai et al., 2024; Zhu et al., 2024)

[1]School of Computer Science and Engineering, University of Electronic Science and Technology of China, Chengdu, China [2]Sun Yat-sen University, Guangzhou, China [3]North China Institute of Computer Systems Engineering. Correspondence to: Shuhang Gu <shuhanggu@gmail.com>.

*Proceedings of the 42$^{st}$ International Conference on Machine Learning*, Vancouver, Canada. PMLR 267, 2025. Copyright 2025 by the author(s).

adopts the spectrum of the Neural Tangent Kernel (NTK) matrix (Jacot et al., 2018) to analyze spectral bias. Taking a step further, Geifman et al. (2023) have made a first attempt to modify the spectrum of the NTK matrix to improve convergence speeds, but limited to the toy MLP and restricted synthetic data. These prior works imply the potential of the NTK matrix to guide adjustments in training dynamics for better INRs. However, the NTK matrix is known for the general lack of an analytical expression and the quadratic growth in memory consumption as the data size increases (Xu et al., 2021; Novak et al., 2022; Mohamadi et al., 2023), especially in INRs, which involve various activation functions and dense predictions. Thus, how to effectively adjust training dynamics of MLPs to purposefully overcome spectral bias in INRs remains an open and valuable problem.

In this paper, we delve into the linear dynamics model of MLPs and propose an effective training dynamics adjustment strategy guided by theoretical derivation to purposefully alleviate the spectral bias issue. Our strategy allows for tailored impacts on spectral bias, achieving more stable improvements. Specifically, these impacts on spectral bias are tailored by purposefully adjusting the spectrum of the NTK matrix, which closely connects the spectral bias with training dynamics. Given that the NTK matrix is not available in most cases, we theoretically justify the empirical NTK (eNTK) matrix (Novak et al., 2022) as a tractable surrogate. Empirical evidence corroborates these theoretical results. Based on this insight, we further propose an inductive gradient adjustment (IGA) method, which could purposefully improve the spectral bias via inductive generalization of the eNTK-based gradient transformation matrix with millions of data points. We give both theoretical and empirical analysis of our method, demonstrating how it tailors impacts on spectral bias. Besides, our IGA method can work with previous structural improvement methods such as Positional Encoding (Tancik et al., 2020) and periodic activations (Sitzmann et al., 2020). We validate our method in various vision applications of INRs and compare it to previous training dynamics adjustment methods. Experimental results show the superiority of our IGA method. Our contributions are summarized as follows:

- We connect spectral bias with linear dynamics model of MLPs and derive a training dynamics adjustment strategy guided by a theoretical standpoint to purposefully improve the spectral bias.

- We propose a practical inductive gradient adjustment method which could purposefully mitigate spectral bias of MLPs even with millions of data points. Theoretical and empirical analyses show how our method tailors impacts on spectral bias.

- We present comprehensive experimental analyses on synthetic data and a range of implicit neural representation tasks. Compared to existing training dynamics approaches, our method enables tailored improvements in mitigating the spectral bias of standard MLPs, resulting in implicit neural representations with enhanced high-frequency details.

## 2. Related Work

**Implicit neural representations.** Implicit neural representations (INRs) representing the discrete signal as an implicit continuous function by MLPs have gained lots of attention. Comparing to traditional grid-based methods, INRs offer remarkable accuracy and memory efficiency across various representation tasks such as 1D audio representation (Kim et al., 2022), 2D image representation (Klocek et al., 2019; Strümpler et al., 2022), 3D shape representation (Park et al., 2019; Martel et al., 2021), novel view synthesis (Mildenhall et al., 2021; Saratchandran et al., 2024). However, vanilla MLPs with ReLU activations fail to represent complex signals. Therefore, various modifications have been studied. One category focuses on input embedding, such as using Positional Encoding (PE) (Mildenhall et al., 2021) or learned features (Takikawa et al., 2021; Martel et al., 2021; Xie et al., 2023). A different category of modifications concentrates on better activations; for instance, periodic activations (Sitzmann et al., 2020; Fathony et al., 2020; Zhu et al., 2024) and Gabor wavelet activations (Saragadam et al., 2023) provide more accurate and robust representations. Considering a broader framework, these works can be summarized as efforts to enhance the representational capacity of INRs or MLPs (Yüce et al., 2022). Recently, another category, distinct from those previously discussed, focuses on the training process of MLPs. Shi et al. (2024a) find that learning parameters in the Fourier domain, i.e., Fourier reparameterized training (FR), can improve approximation accuracy of INRs without altering the inference structure. Cai et al. (2024) propose that the classic batch normalization layer (BN) can also improve the performance of INRs.

**Spectral bias.** Spectral bias refers to the learning bias of MLPs to prioritize learning simple patterns or low-frequency components of the target function. Great efforts have been made to dissect this bias. Rahaman et al. (2019) find that spectral bias varies with the width, depth and input manifold of MLPs. Xu (2018) attribute this bias of MLPs with Tanh activation to the uneven distribution of gradients in the frequency domain. From the perspective of linear dynamics model of MLPs, spectral bias is induced by the uneven spectrum of the corresponding Neural Tangent Kernel (NTK) matrix (Jacot et al., 2018; Arora et al., 2019; Ronen et al., 2019). Inspired by this theoretical result, a series of works (Tancik et al., 2020; Shi et al., 2024a; Cai et al., 2024) try to elucidate spectral bias by observing spectral distribution of

the NTK matrix and find that PE, FR and BN help to attain more even spectrum. In this paper, we adopt the insight of attaining more even spectrum of the NTK matrix. For the first time, we show that modifying the eNTK matrix spectrum can have the similar impacts on spectral bias and propose a practical gradient adjustment method, which improves spectral bias of INRs via inductive generalization of eNTK-based gradient transformation matrix.

## 3. Method

In this section, we first review the linear dynamics model of MLPs, indicating the potential of adjusting the NTK spectrum to overcome spectral bias in INRs. Then we analyze the NTK-based adjustment and show that the NTK matrix's intractability renders it impractical. By theoretical analysis, we justify the eNTK matrix as a tractable proxy, but it succumbs to dimensionality challenges as data size grows. Thus, we propose a practical gradient adjustment method that improves spectral bias via inductive generalization of eNTK-based gradient transformation matrix.

### 3.1. Connecting spectral bias with training dynamics

Without loss of generality and for simplicity, we consider a scalar discrete signal. We define the vector of signal values as $\boldsymbol{y} = (y_1, \ldots, y_N) \in \mathbb{R}^N$, and the corresponding coordinate vectors $(\boldsymbol{x}_1, \ldots, \boldsymbol{x}_N)^\top$, denoted by $\boldsymbol{X} \in \mathbb{R}^{N \times d}$. The INR of this signal is parameterized as $f(\boldsymbol{X}; \Theta)$, where $f(\cdot; \Theta)$ denotes an MLP. Parameters $\Theta \in \mathbb{R}^q$ are optimized with respect to $\frac{1}{2} \sum_{i=1}^N (f(\boldsymbol{x}_i; \Theta) - y_i)^2$ via gradient descent. The residual $(f(\boldsymbol{x}_i; \Theta_t) - y_i)_{i=1}^N$ at time step $t$ is denoted as $\boldsymbol{r}_t$. Training dynamics of $\boldsymbol{r}_t$ can be approximated by the linear dynamics model: $\boldsymbol{r}_t = (\boldsymbol{I} - \eta \boldsymbol{K}) \boldsymbol{r}_{t-1}$ under the conditions of wide networks and small learning rates $\eta$ (Du et al., 2018; Arora et al., 2019; Lee et al., 2019). $\boldsymbol{K}$ is the NTK matrix defined by $\mathbb{E}_{\Theta_0}[\nabla_\Theta f(\boldsymbol{X}; \Theta_0)^\top \nabla_\Theta f(\boldsymbol{X}; \Theta_0)]$, where $\nabla_\Theta f(\boldsymbol{X}; \Theta_0) \in \mathbb{R}^{q \times N}$. With the eigenvalue decomposition of $\boldsymbol{K} = \sum_{i=1}^N \lambda_i \boldsymbol{v}_i \boldsymbol{v}_i^\top$ and recurrence unfolding of the linear dynamics model, training dynamics of $\boldsymbol{r}_t$ are characterized as follows (Arora et al., 2019):

$$\|\mathbf{r}_t\|_2 = \sqrt{\sum_{i=1}^N (1 - \eta \lambda_i)^{2t} (\boldsymbol{v}_i^\top \boldsymbol{y})^2}. \tag{1}$$

Eq. 1 shows that convergence rate of $f(\boldsymbol{x}, \Theta)$ at the projection direction $\boldsymbol{v}_i^\top$ with larger $\lambda_i$ is faster. For vanilla MLPs, projection directions related to high frequencies are consistently assigned to small eigenvalues, while those related to low frequencies correspond to larger eigenvalues (Ronen et al., 2019; Bietti & Mairal, 2019; Heckel & Soltanolkotabi, 2020). Such an uneven spectrum of $\boldsymbol{K}$ leads to extremely slow convergence rates to high frequency components of the signal, leading to a reduction in average approximation accuracy (e.g., MSE loss) as well as frequency-aware met-

rics (e.g., SSIM, MS-SSIM (Wang et al., 2003)) in INRs. Following this insight, recent works (Tancik et al., 2020; Bai et al., 2023; Shi et al., 2024a; Cai et al., 2024) suggest that MLPs with a more even spectrum of $\boldsymbol{K}$ are less affected by spectral bias.

### 3.2. Inductive Gradient Adjustment

As previously discussed, such an uneven spectrum of $\boldsymbol{K}$ leads to slow convergence of high frequency components. Naturally, adjusting the evenness of the spectrum of $\boldsymbol{K}$ is expected to tailor the improvement of spectral bias. Following this intuition, we focus on the adjustment of eigenvalues of $\boldsymbol{K}$, i.e., $\boldsymbol{K}$-based adjustment. Inspired by the theoretical insights of Geifman et al. (2023), this can be achieved by multiplying a transformation matrix $\boldsymbol{S}$ as follows[1]:

$$\Theta_{t+1} = \Theta_t - \eta \nabla_\Theta f(\boldsymbol{X}; \Theta) \boldsymbol{S} \boldsymbol{r}_t, \tag{2}$$

where $\boldsymbol{S} = \sum_{i=1}^N (g_i(\lambda_i)/\lambda_i) \boldsymbol{v}_i \boldsymbol{v}_i^\top$; $\{g_i\}_{i=1}^N$ denote the potential eigenvalue transformations. As driven by the training dynamics of Eq. 2, the spectrum of the modified kernel matrix $\boldsymbol{K}$ can be theoretically approximated as $\{g_i(\lambda_i)\}_{i=1}^N$.

Although Eq. 2 offers a theoretical approach to adjust impacts of training dynamics on spectral bias, Eq. 2 is almost infeasible in practical INRs tasks. The first challenge is that an analytical expression for $\boldsymbol{K}$ is difficult to derive (Xu et al., 2021; Novak et al., 2022; Wang et al., 2023; Shi et al., 2024b). Specifically, the analytical expression $\mathbb{E}_{\Theta_0}[\nabla_\Theta f(\boldsymbol{X}; \Theta_0)^\top \nabla_\Theta f(\boldsymbol{X}; \Theta_0)]$ of $\boldsymbol{K}$ becomes intractable as the depth of the network increases. The second challenge is that the size of $\boldsymbol{K}$ grows quadratically with the number of data points (Mohamadi et al., 2023). Considering the Kodak image fitting task (Saragadam et al., 2023; Shi et al., 2024a), it requires decomposing a matrix on the order of $10^{11}$ entries, which occupies over $8192$GiB (gibibytes) of memory if stored in double precision.

To overcome the first challenge, numerous studies (Xu et al., 2021; Novak et al., 2022; Mohamadi et al., 2023) have studied the eNTK matrix $\tilde{\boldsymbol{K}}$ defined by $\nabla_{\Theta_t} f(\boldsymbol{X}; \Theta_t)^\top \nabla_{\Theta_t} f(\boldsymbol{X}; \Theta_t)$ and Geifman et al. (2023) show that replacing $\boldsymbol{K}$ with $\tilde{\boldsymbol{K}}$ ($\tilde{\boldsymbol{K}}$-based) in Eq. 2 can accelerate convergence on toy settings. However, whether $\tilde{\boldsymbol{K}}$-based adjustment has the similar impacts on spectral bias as in Eq. 2 and remains consistently effective in general settings, is an open question. To justify feasibility, we prove that eigenvectors of $\tilde{\boldsymbol{K}}$ as well as eigenvalues converge to the corresponding parts of $\boldsymbol{K}$ as the network width increases. Namely, impacts on spectral bias by $\tilde{\boldsymbol{K}}$-based adjustment can be approximately equivalent to that of $\boldsymbol{K}$-based adjustment. The detailed theoretical analysis is presented in our Theorem 3.1. Then, we empirically validate the aforementioned equivalence and demonstrate that

---

[1]This matrix is also known as the pre-conditioned matrix in the field of optimization.

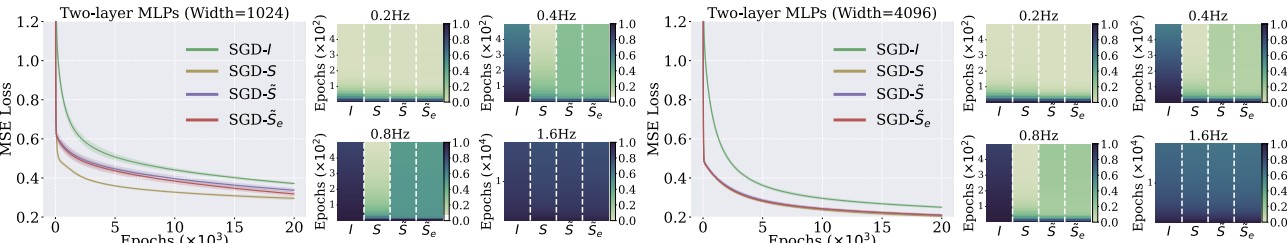

*Figure 1.* Evolution of approximation error with training iterations in time and Fourier domain. Line plots visualize MSE loss curves of MLPs with 1024 and 4096 neurons optimized by four gradient adjustments, i.e., $I, S, \tilde{S}, \tilde{S}_e$. The shaded area of lines indicates training fluctuations. Heatmaps show the relative error $\Delta_k$ in Eq. 6 on four frequency bands. Details are in the experiment 1 of Sec. 4.

$\tilde{K}$-based adjustment can tailor the improvement of spectral bias on more general settings in Sec. 4. The formula of $\tilde{K}$-based adjustment is as follows:

$$\Theta_{t+1} = \Theta_t - \eta \nabla_{\Theta_t} f(\boldsymbol{X}; \Theta_t) \tilde{\boldsymbol{S}} \boldsymbol{r}_t, \qquad (3)$$

where $\tilde{\boldsymbol{K}} = \sum_{i=1}^N \tilde{\lambda}_i \tilde{\boldsymbol{v}}_i \tilde{\boldsymbol{v}}_i^\top$; $\tilde{\boldsymbol{S}} = \sum_{i=1}^N (g_i(\tilde{\lambda}_i)/\tilde{\lambda}_i) \tilde{\boldsymbol{v}}_i \tilde{\boldsymbol{v}}_i^\top$.

Although adjustment based on $\tilde{K}$ instead of $K$ achieves similar effects to $K$ while avoiding intractability, it still encounters the curse of dimensionality as the data size $N$ increases. One common approach is to estimate $\tilde{K}$ using sampled data, which is efficient for tasks where $\tilde{K}$ indirectly impacts training without affecting gradients, such as neural architecture search (Park et al., 2020) or adaptive module (Shi et al., 2024b). However, how to effectively utilize $\tilde{K}$ from sampled data to directly adjust the training dynamics on the population data remains a problem. In Theorem 3.2, we show that training dynamics of $\boldsymbol{r}_t$ could be estimated via inductive generalization of the linear dynamics model of sampled data and analyzable error. Inspired by this theoretical foundation, we propose to inductively generalize the gradient adjustment based on sampled data to the gradients of the population data. Specifically, considering an input set $\boldsymbol{X} = \{\boldsymbol{x}_i\}_{i=1}^N$ of $N$ coordinates in $\mathbb{R}^d$ as the population data, $\boldsymbol{X}$ can be divided into $n$ groups ($N = np$), where the $j$-th group $\boldsymbol{X}^j$ and corresponding labels are $\{(\boldsymbol{x}_i^j, y_i^j)\}_{i=1}^p$. Then we sample one data point from each group to form the sampled set $\boldsymbol{X}_e$, where $|\boldsymbol{X}_e| = n \ll N$. The eNTK matrix on $\boldsymbol{X}_e$ is defined by $\tilde{\boldsymbol{K}}_e = \nabla_{\Theta_t} f(\boldsymbol{X}_e; \Theta_t)^\top \nabla_{\Theta_t} f(\boldsymbol{X}_e; \Theta_t)$. We construct the corresponding transformation matrix $\tilde{\boldsymbol{S}}_e$. Then inductive adjustments $\tilde{\boldsymbol{S}}_e$ are generalized to gradients of the population data $\boldsymbol{X}$ as follows:

$$\Theta_{t+1} = \Theta_t - \eta \sum_{i=1}^p \nabla_{\Theta_t} f(\boldsymbol{X}_i, \Theta_t) \tilde{\boldsymbol{S}}_e \boldsymbol{r}_t^i, \qquad (4)$$

where $\boldsymbol{r}_t^i$ is the residual of $\boldsymbol{X}_i = \{(\boldsymbol{x}_i^j, y_i^j)\}_{j=1}^n$; $\tilde{\boldsymbol{S}}_e = \sum_{i=1}^n (g_i(\tilde{\tilde{\lambda}}_i)/\tilde{\tilde{\lambda}}_i) \tilde{\tilde{\boldsymbol{v}}}_i \tilde{\tilde{\boldsymbol{v}}}_i^\top$; $\tilde{\boldsymbol{K}}_e = \sum_{i=1}^n \tilde{\tilde{\lambda}}_i \tilde{\tilde{\boldsymbol{v}}}_i \tilde{\tilde{\boldsymbol{v}}}_i^\top$. We introduce the implementation details of Eq. 4 in Sec. 3.4 and summarize the IGA method in the form of an algorithm block 1.

As shown in equation 4, all $\boldsymbol{r}_t^i$ are linearly transformed by $\tilde{\boldsymbol{S}}_e$. Please note that for each vector $\boldsymbol{r}_t^i$ (where $i = 1, \ldots, p$), its $j$-th element is scaled by the $j$-th column of the matrix $\tilde{\boldsymbol{S}}_e$.

---

**Algorithm 1** Inductive Gradient Adjustment (IGA)

**Input:** Mini-batch $X_t = \{(x_i, y_i)\}_{i=1}^N$; model $f(\cdot, \Theta_t)$
**Output:** Updated parameters $\Theta_{t+1}$
**// Sample a subset**
Sample $X_e \subset X_t$ based on the strategy in Sec. 3.4
**// Compute gradients and empirical kernel**
Compute $G_e = \nabla_{\Theta_t} f(X_e; \Theta_t)$ and then $\tilde{K}_e = G_e^\top G_e$
**// Construct the transformation matrix**
Compute $\tilde{S}_e$ based on $\tilde{K}_e$ using the method in Sec. 3.4
**// Generalize the adjustment to the full mini-batch**
Adjusted gradient: $\tilde{g}_t = \sum_{i=1}^p \nabla_{\Theta_t} f(\boldsymbol{X}_i, \Theta_t) \tilde{S}_e \boldsymbol{r}_t^i$; update the parameters by: $\Theta_{t+1} \leftarrow \Theta_t - \eta \cdot \tilde{g}_t$
**return** $\Theta_{t+1}$

---

And the $j$-th elements of these vectors $\boldsymbol{r}_t^i$ all belong to the group $\boldsymbol{X}^j$. Therefore, our inductive gradient adjustments are generalized from one sampled data point of $\boldsymbol{X}^j$ to the remaining $(p-1)$ data points of $\boldsymbol{X}^j$, for all $j = 1, \ldots, n$.

### 3.3. Theoretical Analysis

In this subsection, we provide theoretical analysis to justify the feasibility of our IGA. In Theorem 3.1, we prove that adjustments based on eNTK matrix $\tilde{K}$ asymptotically converge to those based on NTK matrix $K$ as the network width increases. Then we delve into the estimate of training dynamics via inductive generalization of linear dynamics model of sampled data in Theorem 3.2.

In specific, we build upon previous training dynamics works (Du et al., 2018; Arora et al., 2019; Lee et al., 2019; Geifman et al., 2023) and analyze a two-layer network $f(\boldsymbol{x}; \Theta)$ with width $m$. Without loss of generality, we assume the training set $\{\boldsymbol{x}_i, y_i\}_{i=1}^N$ and adopt the notations in Eq. 2-4. Detailed analysis settings are in Appendix A. Based on these settings, the following Theorem 3.1 show that $\tilde{K}$-based adjustments achieves similar effects to $K$-based adjustments.

**Theorem 3.1.** *(informal) Assuming that* $\boldsymbol{v}_i^\top \tilde{\boldsymbol{v}}_i > 0$ *for* $i \in [N]$. *Let* $\{g_i(x)\}_{i=1}^N$ *be a set of Lipschitz continuous functions, with learning rate* $\eta < \min\{(\max(g_i(\lambda)) + \min(g_i(\lambda)))^{-1}, (\max(g_i(\tilde{\lambda})) + \min(g_i(\tilde{\lambda})))^{-1}\}$, *for all* $\epsilon > 0$, *there always exists* $M > 0$, *such that* $m > M$, *for* $i \in [N]$, *we have:* $|g_i(\lambda_i) - g_i(\tilde{\lambda}_i)| < \epsilon_1, \|\boldsymbol{v}_i - \tilde{\boldsymbol{v}}_i\| < \epsilon_2,$

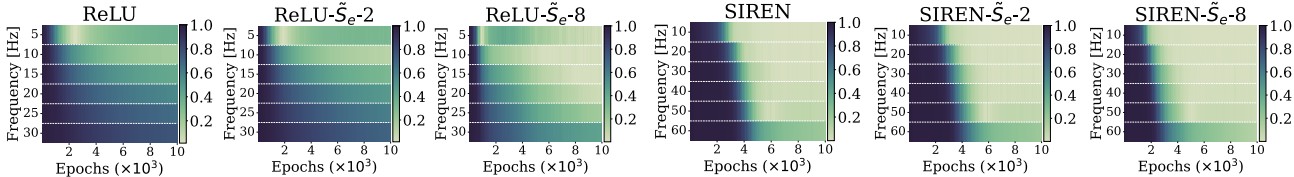

*Figure 2.* Progressively amplified impacts on spectral bias of ReLU and SIREN by increasing the number of balanced eigenvalues via $\tilde{S}_e$ when $p = 8$, i.e., IGA. ReLU denotes the MLP with ReLU optimized via vanilla gradients; ReLU-$\tilde{S}_e$-2 denotes the ReLU optimized via gradients adjusted by $\tilde{S}_e$ with 2 balanced eigenvalues. Details and more results are in the experiment 2 of Sec. 4 and Appendix B.

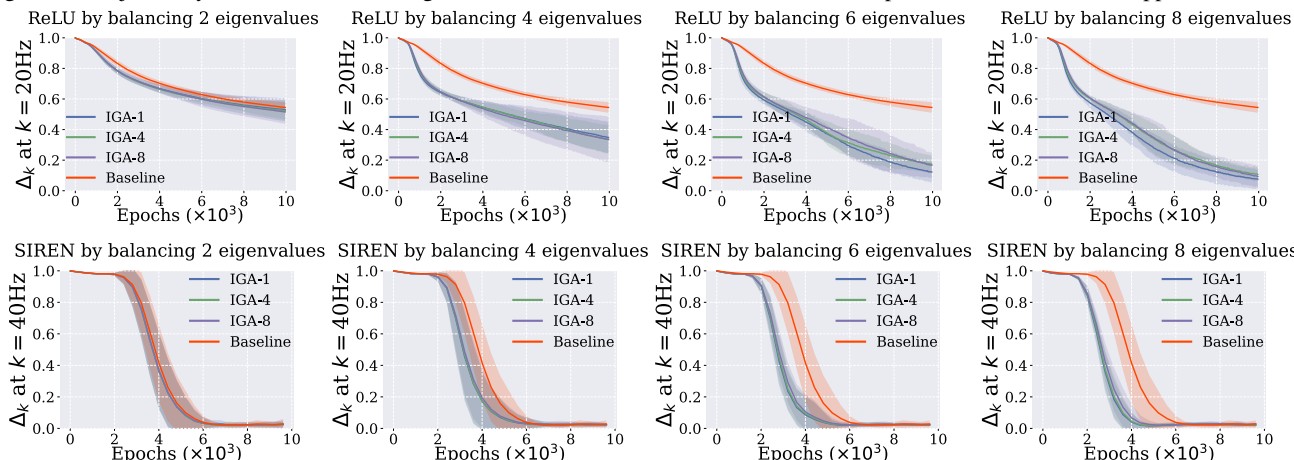

*Figure 3.* Relative error $\Delta_k$ curves of ReLU at 20Hz and SIREN at 40Hz with varying balanced eigenvalues and group size $p$. IGA-1 denotes that $p = 1$, i.e., $\tilde{K}$-based gradient adjustment; Baseline denotes vanilla gradients. The shaded area of lines indicates training fluctuations, represented by twice the standard deviation. Details and more results are in the experiment 2 of Sec. 4 and Appendix B.

thus

$$|(1 - \eta g_i(\tilde{\lambda}_i))^2 (\tilde{v}_i^\top r_t)^2 - (1 - \eta g_i(\lambda_i))^2 (v_i^\top r_t)^2| < \epsilon_3,$$

where $\epsilon_1, \epsilon_2, \epsilon_3$ and $\epsilon$ are of the same order as $\epsilon \to 0$.

Theorem 3.1 illustrates that eigenvalues and eigenvectors of $\tilde{K}$ converge one-to-one to eigenvalues and eigenvectors of $K$. That is, $\tilde{S}$ impacts not only the convergence rates $\{g_i(\tilde{\lambda}_i)\}_{i=1}^N$ similar to those of $S$ but also the corresponding convergence directions ($\tilde{v}_i$). This theoretical results justify that the $\tilde{K}$-based adjustment is approximately equivalent to $K$-based adjustment. We further provide the empirical evidence in Sec. 4 and detailed proof of formal version in our Appendix A. Next, we analyze how training dynamics of $X$ can be estimated from sampled data $X_e$.

**Theorem 3.2.** *(informal)* $X$ *is partitioned into* $n$ *groups* $X_j = \{x_1^j, \ldots, x_p^j\}$ *and* $N = np$. *For* $X_j$, *the sampled point denoted as* $x^j$ *and these* $n$ *points form* $X_e = \{x^j\}_{j=1}^n, n \ll N$. *There exists* $\epsilon > 0$, *for* $j \in [n]$ *and* $i_1, i_2 \in [p]$, *such that* $\max\{|(f(x_{i_1}^j, \Theta_t) - y_{i_1}^j) - (f(x_{i_2}^j, \Theta_t) - y_{i_2}^j)|, \|\nabla_{\Theta_t} f(x_{i_1}^j) - \nabla_{\Theta_t} f(x_{i_2}^j)\|\} < \epsilon$. *Then, for any* $x_i$, *assumed that* $x_i$ *belongs to* $X_{j_i}$, *such that:*

$$|(\Delta r_t)_i - p \nabla_{\Theta_t} f(x^{j_i})^\top \nabla_{\Theta_t} f(X_e; \Theta_t) r_e^t| < \epsilon_1 + \epsilon_2,$$

*where* $\epsilon_1 = \Theta(\epsilon \frac{n^{3/2}}{\sqrt{m}})$; $\epsilon_2 = \Theta(m^{-3/2})$; $\Theta(\cdot)$ *denotes equivalence;* $(\Delta r_t)_i = (r_{t+1} - r_t)_i$ *denotes the residual*

*of* $x_i$ *at time step* $t$; $r_e^t$ *denotes the residual on* $X_e$ *at time step* $t$; $\nabla_{\Theta_t} f(x^{j_i})^\top \nabla_{\Theta_t} f(X_e; \Theta_t)$ *is the* $j_i$-*th row in* $\tilde{K}_e$.

Theorem 3.2 demonstrates that generalizing $\tilde{K}_e$, induced from sampled data $X_e$, to population data $X$ provides an inductive estimation of overall training dynamics. The inductive estimate error $\epsilon_1 + \epsilon_2$ decreases as the network width $m$ increases, the consistency bound $\epsilon$ decreases, or both. This theoretical insight encourages us to generalize the transformation matrix $\tilde{S}_e$ to gradients of population data $X$, thereby tailoring impacts on spectral bias via $K_e, \tilde{S}_e$. We provide empirical validation of Theorem 3.2 in Sec. 4, with its formal version and proof in Appendix A.

### 3.4. Implementation Details

**Sampling Strategy.** As illustrated in Sec. 3.2, we need to sample $X_e$ from groups of the population data $X$ to obtain $\tilde{K}_e, \tilde{S}_e$. In practice, we first form $n$ groups by simply considering $p$ adjacent input coordinates for each group. Specifically, for 1D and 2D inputs, groups are defined by non-overlapping intervals and patches, respectively. For $N$ $d$-dimensional input coordinates ($d \geq 3$), typically represented as a tensor of shape $(n_1, \ldots, n_d, d)$, we flatten the first $d$ dimensions into a 2D tensor $X'$ of shape $(N, d)$ and then group it along the first dimension of $X'$ as 1D signals. Then, the input with the largest residuals in each group are selected to form the sampled data $X_e$. To better illustrate the above process, we provide diagrams in Fig. 6. Details

of hyperparameters $n$ and $p$ will be introduced in the experiment section. The ablation study of $n$ and $p$ is in the Appendix F. As shown in our wide experiments, the above strategy is simple yet effective and easy to implement.

**Construction Strategy.** As illustrated in Eq. 4 and Theorem 3.2, we adjust the evenness of the spectrum of $\tilde{\boldsymbol{K}}_e$ via the transformation matrix $\tilde{\boldsymbol{S}}_e$ to purposefully improve the spectral bias. We have $\tilde{\boldsymbol{K}}_e = \sum_{i=1}^{n} \tilde{\tilde{\lambda}}_i \tilde{\boldsymbol{v}}_i \tilde{\boldsymbol{v}}_i^\top$ and assume that $\tilde{\tilde{\lambda}}_1 > \cdots > \tilde{\tilde{\lambda}}_n > 0$. To adjust the evenness of spectrum of $\tilde{\boldsymbol{K}}_e$, $\tilde{\boldsymbol{S}}_e$ is constructed to balance the eigenvalues of different eigenvectors of $\tilde{\boldsymbol{K}}_e$; impacts are tailored by managing the number of balanced eigenvalues. For vanilla gradient descent, $\tilde{\boldsymbol{S}}_e$ is constructed as follows:

$$\tilde{\boldsymbol{S}}_e = \sum_{i=start}^{end} (\tilde{\tilde{\lambda}}_{start}/\tilde{\tilde{\lambda}}_i)\tilde{\mathbf{v}}_i\tilde{\mathbf{v}}_i^\top + \sum_{i\notin[start,end]} 1\cdot\tilde{\mathbf{v}}_i\tilde{\mathbf{v}}_i^\top. \quad (5)$$

Generally, $start$ is fixed at 1; $end$ represents the controlled spectral range. Larger $end$ values have stronger impacts on spectral bias. We demonstrate this effect in experiment 2 of Sec. 4. For Adam, we utilize $\tilde{\tilde{\lambda}}_{end+1}$ instead of $\tilde{\tilde{\lambda}}_{start}$ to balance eigenvalues in Eq. 5 for better convergence, resulting in uniform eigenvalues of the modified $\tilde{\boldsymbol{K}}_e$ across the $(end - start + 2)$ eigenvectors. Then, momentum and adaptive learning rates are updated according to the adjusted gradients. More discussion are in Appendix D.

**Multidimensional Output Approximation.** In practical INRs tasks, the output is multidimensional, such as for color images. Compared to the data size $N$, it is trivial but still affects efficiency. Thus, we simply compute the Jacobian matrix $\nabla_\Theta f(\boldsymbol{X};\Theta)$ by sum-of-logits, which is shown to be efficient in Mohamadi et al. (2023); Shi et al. (2024b).

# 4. Empirical Analysis on Simple Function Approximation

In this section, we aim to explore and provide empirical evidence for the theoretical results presented in Sec. 3.3. In experiment 1, we investigate the effects of $\boldsymbol{K}$-based and $\tilde{\boldsymbol{K}}$-based adjustments on spectral bias and observe that their differences diminish as the network width $m$ increases (Theorem 3.1). Training dynamics curves further suggest that increasing width $m$ reduces estimate error $\epsilon_1 + \epsilon_2$ from inductive generalization (Theorem 3.2). In experiment 2, we validate $\tilde{\boldsymbol{K}}$-based adjustments and IGA on more general settings, demonstrating that IGA purposefully improves spectral bias, with its impacts controllable by managing the number of balanced eigenvalues in Eq. 5. This suggests the effectiveness of inductive generalization in estimating the training dynamics (Theorem 3.2) on general settings.

To better analyze, we follow Xu (2018); Shi et al. (2024a) and compute the relative error $\Delta_k$ between target signal $f$

and outputs $f_\Theta$ at frequency $k$ to show spectral bias:

$$\Delta_k = |\mathcal{F}_D[f](k) - \mathcal{F}_D[f_\Theta](k)|/|\mathcal{F}_D[f](k)|, \quad (6)$$

where $\mathcal{F}_D$ denotes the discrete Fourier transform.

**Experiment 1.** In this experiment, as discussed above, we investigate the $\boldsymbol{K}$-based and $\tilde{\boldsymbol{K}}$-based adjustments. Therefore, the accurate computation of NTK matrix $\boldsymbol{K}$ is the key. Given that the accurate computation of $\boldsymbol{K}$ is non-trivial, we adopt a two-layer MLP with a fixed last layer as in Arora et al. (2019); Ronen et al. (2019), whose $\boldsymbol{K}_{ij}$ can be computed by the formula $\frac{1}{4\pi}(\mathbf{x}_i^\top\mathbf{x}_j + 1)(\pi - \arccos(\mathbf{x}_i^\top\mathbf{x}_j))$. This architecture facilitates theoretical analysis but has limited representational capacity for complex functions. Considering these concerns, we constructed the following simple function $f(\cos(\theta), \sin(\theta)) : \mathbb{S}^1 \to \mathbb{R}^1$, widely used in prior works (Ronen et al., 2019; Geifman et al., 2023):

$$\sin(0.4\pi\theta) + \sin(0.8\pi\theta) + \sin(1.6\pi\theta) + \sin(3.2\pi\theta). \quad (7)$$

We set $N = 1024$ for input samples via uniformly sampling 1024 $\theta$ in $[0, 2\pi]$. We vary the width of the MLP from 1024 to 8192. We adopt the Identity matrix $\boldsymbol{I}$ (that is, the vanilla gradient) and a series of $\boldsymbol{S}, \tilde{\boldsymbol{S}}, \tilde{\boldsymbol{S}}_e$ that $start = 1$ and $end$ ranges from 10 to 14. Following our sampling strategy, we set $p = 8$ and $n = 128$, indicating that $\tilde{\boldsymbol{S}}_e$ is only $1/64$ the size of $\boldsymbol{S}, \tilde{\boldsymbol{S}}$. All MLPs are trained separately with the same fixed learning rates by SGD for 20K iterations. All models are repeated by 10 random seeds. We visualize the results of the baseline model and the adjustments with $end = 14$ in Fig. 1. More results can be found in Appendix B.

**Analysis of Experiment 1.** In Fig. 1, line plots show the convergence trends of MLPs by $\boldsymbol{K}$-based ($\boldsymbol{S}$), $\tilde{\boldsymbol{K}}$-based ($\tilde{\boldsymbol{S}}$) and inductive gradient ($\tilde{\boldsymbol{S}}_e$) adjustments, compared to vanilla gradient ($\boldsymbol{I}$). Heat maps with a consistent color scale visualize their spectral impacts, where darker colors indicate higher errors. Note that color bars of $\boldsymbol{I}$ darken rapidly from 0.4Hz, highlighting severe spectral bias in MLPs trained with vanilla gradients. All the adjustments introduce lighter shades over 0.4Hz and 0.8Hz, effectively improving spectral bias, therefore resulting in faster MSE convergence. Please note that while adjustments by $\boldsymbol{S}$, $\tilde{\boldsymbol{S}}$ and $\tilde{\boldsymbol{S}}_e$ exhibit slight difference in MSE loss trends and impacts on spectrum in the case of width 1024, the differences are barely noticeable for width 4096. This result aligns with previous theoretical analysis that increasing width $m$ reduces differences between $\boldsymbol{K}$-based and $\tilde{\boldsymbol{K}}$-based adjustments (Theorem 3.1) and errors induced by inductive generalization (Theorem 3.2). Moreover, despite the presence of difference, $\tilde{\boldsymbol{S}}$ and $\tilde{\boldsymbol{S}}_e$ effectively improve spectral bias and the approximation accuracy, indicating that $\tilde{\boldsymbol{K}}$ successfully links spectral bias with training dynamics and our IGA method is effective.

**Experiment 2.** In this experiment, we aim to validate $\tilde{\boldsymbol{K}}$-based adjustments and our IGA on more general settings. For IGA, we also aim to show that impacts on spectral

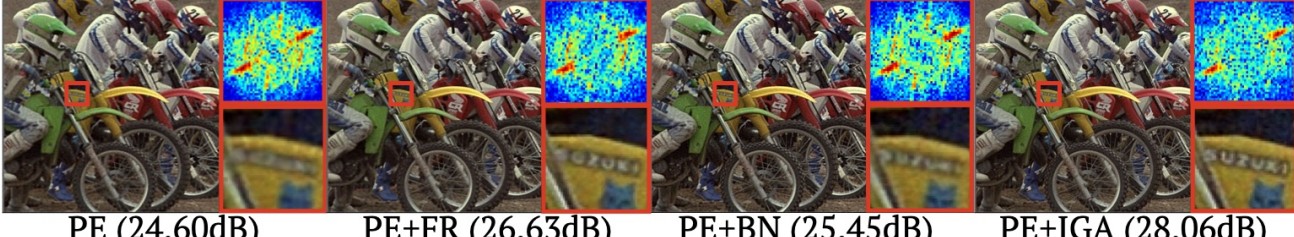

| PE (24.60dB) | PE+FR (26.63dB) | PE+BN (25.45dB) | PE+IGA (28.06dB) |

*Figure 4.* Visual examples of 2D color image approximation results by different training dynamics methods. Enlarged views of regions labeled by red boxes are provided. The residuals of these regions in the Fourier domain are visualized through heatmaps in the top right corner. The increase in error corresponds to the transition of colors in the heatmaps from blue to red. Detailed settings are in Sec. 5.1.

*Table 1.* Average metrics of 2D color image approximation results by different methods. The detailed settings can be found in Sec. 5.1. Per-scene results are provided in Table 8-11. Note: SIREN+BN is omitted due to our finding of their incompatibility, neither does the original work (Cai et al., 2024) demonstrate the feasibility.

| Average Metric | ReLU | | | | PE | | | | SIREN | | |
|---|---|---|---|---|---|---|---|---|---|---|---|
| | Vanilla | +FR | +BN | **+IGA** | Vanilla | +FR | +BN | **+IGA** | Vanilla | +FR | **+IGA** |
| PSNR ↑ | 21.78 | 22.14 | 22.55 | **23.00** | 28.64 | 31.65 | 28.78 | **32.46** | 32.65 | 32.61 | **33.48** |
| SSIM ↑ | 0.4833 | 0.4919 | 0.5004 | **0.5126** | 0.7832 | 0.8167 | 0.8030 | **0.8822** | 0.8975 | 0.8991 | **0.9121** |
| MS-SSIM ↑ | 0.6521 | 0.6800 | 0.7090 | **0.7383** | 0.9466 | 0.9564 | 0.9554 | **0.9752** | 0.9818 | 0.9820 | **0.9847** |
| LPIPS ↓ | 0.6302 | 0.6315 | 0.6182 | **0.5549** | 0.2223 | 0.1869 | 0.2346 | **0.0938** | 0.0807 | 0.0813 | **0.0668** |

bias can be tailored by managing the number of balanced eigenvalues under varying $p$. Therefore, we consider two practical INR models, i.e., MLPs with ReLU (ReLU) and MLPs with Sine (SIREN) (Sitzmann et al., 2020) like Shi et al. (2024a). To analyze tailored impacts under varying $p$, we use settings in Rahaman et al. (2019) and construct a 1D function $f : \mathbb{R}^1 \to \mathbb{R}^1$ with abundant spectrum as follows:

$$\begin{aligned} \sin(20\pi x) + \sin(40\pi x) + \sin(60\pi x) + \sin(80\pi x) \\ + \sin(100\pi x) + \sin(120\pi x). \end{aligned} \quad (8)$$

SIREN is trained to regress the $f(x)$ with 2048 values uniformly sampled in $[-1, 1]$. For ReLU, we halve the frequency of each component due to its limited representation capacity (Yüce et al., 2022). For IGA, we set $p$ to 1, 4 and 8 and vary $end$ from 1 to 7 to construct the corresponding $\tilde{S}_e$. When $p = 1$, IGA degenerates to $\tilde{K}$-based adjustment. We adopt a four-hidden-layer, 256-width architecture for ReLU and SIREN, optimized by Adam for 10K iterations with a fixed learning rate. All models are repeated by 10 random seeds. In Fig. 2, we adopt heatmaps of relative error $\Delta_k$ evolution to show impacts on spectral bias. In Fig. 3, we visualize $\Delta_k$ curves for some frequencies to precisely demonstrate impacts of IGA on convergence rates.

**Analysis of Experiment 2.** In Fig. 2, for $\tilde{K}$-based adjustment and IGA, as more balanced eigenvalues are introduced, the proportion of lighter regions in heatmaps of both ReLU and SIREN grows, while the relative error curve decreases more rapidly in Fig. 3. These observations clearly illustrate that impacts on spectral bias by $\tilde{K}$-based adjustment and our IGA can be amplified by increasing the number of balanced eigenvalues on general settings. Further, as shown in Fig. 3, although the relative error curves of ReLU exhibit subtle differences as $p$ increases, our IGA effectively amplifies impacts on spectral bias with more balanced eigenvalues. These results demonstrate the consistency between

our theoretical strategy and practical outcomes, supporting the extension of IGA to more general applications.

## 5. Experiment on Vision Applications

In this section, we apply our inductive gradient adjustment (IGA) method to various vision applications of INRs. The superior and consistent improvements demonstrate the broad applicability and superiority of IGA.

### 5.1. 2D Color image approximation

Single natural image fitting has become an ideal test bed for INR models (Saragadam et al., 2023; Xie et al., 2023; Shi et al., 2024a), as natural images contain both low- and high-frequency components (Chan & Shen, 2005). In this experiment, we attempt to parameterize the function $\phi : \mathbb{R}^2 \to \mathbb{R}^3, \mathbf{x} \to \phi(\mathbf{x})$ that represents a discrete image. Following Saragadam et al. (2023), we establish four-layer MLPs with 256 neurons per hidden layer. We test three MLPs architectures, i.e., MLPs with ReLU activations (ReLU), ReLU with Positional Encoding (PE) (Tancik et al., 2020) and MLPs with periodic activation Sine (SIREN) (Sitzmann et al., 2020). These three models are classic baselines in INRs and are widely used for comparison (Sitzmann et al., 2020; Saragadam et al., 2023; Shi et al., 2024a). Results on more activations can be found in Appendix E. We test on the first 8 images from the Kodak 24 (Franzen, 1999), each containing $768 \times 512$ pixels. For these images, $\mathbf{K}$ has approximately $10^{11}$ entries, rendering decomposition and multiplication infeasible during training. Following our sampling strategy, we group each image into non-overlapping $32 \times 32$ patches ($p = 1024$) and sample data with the largest residuals to form the $\mathbf{X}_e$. We construct $\tilde{S}_e$ with $end = 20$ for SIREN and PE and $end = 25$ for ReLU due to its severe spectral bias. Therefore, we generalize the inductive

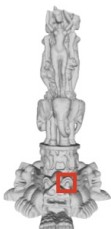 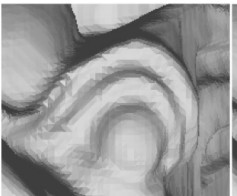 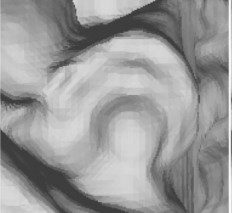 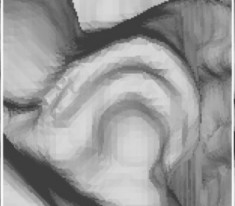 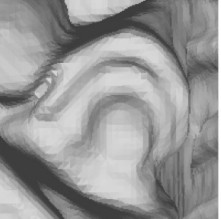 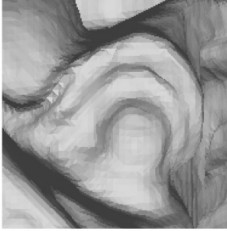

| Thai | Ground truth | PE (0.9897) | PE+FR (0.9929) | PE+BN (0.9916) | PE+IGA (0.9943) |

*Figure 5.* Visual examples of 3D shape representation results by different training dynamics methods. Five images on the right correspond to the enlarged views of the red-boxed area of ground truth and four models, respectively. More results can be found in our Appendix H.

*Table 2.* Intersection over Union (IOU) and Chamfer distance of 3D shape representation by different methods. The detailed settings can be found in Sec. 5.2. Per-scene results are provided in Table 12 and 13.

| Average | ReLU | | | | PE | | | | SIREN | | |
|---|---|---|---|---|---|---|---|---|---|---|---|
| Metric | Vanilla | +FR | +BN | **+IGA** | Vanilla | +FR | +BN | **+IGA** | Vanilla | +FR | **+IGA** |
| IOU ↑ | 9.647e-1 | 9.654e-1 | 9.662e-1 | **9.733e-1** | 9.942e-1 | 9.961e-2 | 9.946e-1 | **9.970e-1** | 9.889e-1 | 9.866e-1 | **9.897e-1** |
| Chamfer Distance ↓ | 5.936e-6 | 5.867e-6 | 5.610e-6 | **5.487e-6** | 5.123e-6 | 5.111e-6 | 5.116e-6 | **5.108e-6** | 5.688e-6 | 5.163e-6 | **5.157e-6** |

gradient adjustments, derived from less than $0.1\%$ of the sampled data, to the entire population. For all architectures, we train baseline with vanilla gradients and with our IGA (+IGA) by Adam optimizer. The learning rate schedule follows Shi et al. (2024a). Detailed training settings can be found in Appendix G. Current training dynamics methods, i.e., Fourier reparameterization (+FR) (Shi et al., 2024a) and batch normalization (+BN) (Cai et al., 2024), are compared. Their hyperparameters follow publicly available codes, and we make every effort to achieve optimal performance. In Table 1, we report average values of four metrics of different INRs over the first 8 images of Kodak 24, where LPIPS values are measured by the "alex" from Zhang et al. (2018).

As shown in Table 1 and Fig. 4, PE with our IGA (PE+IGA) not only achieves the best improvements on average approximation accuracy: PSNR values compared to PE, PE+FR, and PE+BN, but also excels in frequency-aware metrics: SSIM, MS-SSIM, with a more precise representation of high-frequency details instead of being overly smooth like other methods. Concretely, as shown in the spectra located in the top right corner of Fig. 4, improvements of IGA are uniformly distributed across most frequency bands. In contrast, while BN and FR increase PSNR values, most improvements are near the origin, i.e., in the low-frequency range. Furthermore, as shown in Table 8-11, IGA achieves consistent improvements across almost all scenarios and metrics. While FR and BN achieve competitive results in some scenarios, their performance is generally less stable and occasionally leads to suboptimal or negative outcomes. These observations indicate that our IGA method allows for more balanced convergence rates of a wide spectral range with millions of data. More results are in Appendix G.

## 5.2. 3D shape representation

3D shape representation by Signed Distanced Functions (SDFs) is widely used for testing INR models (Saragadam et al., 2023; Cai et al., 2024; Shi et al., 2024a), due to its

ability to model complex topologies. In this section, we evaluate IGA on this task using five 3D objects from the public dataset (Martel et al., 2021; Zhu et al., 2024) and objects are sampled over a $512^3$ grid following Saragadam et al. (2023); Shi et al. (2024a). For IGA, following our sampling strategy for high-dimensional data, we set $p = 256$ and randomly draw 512 groups and generalize the inductive gradient adjustments on these groups in each iteration. The architectures of MLPs are consistent with Experiment 5.1. We use the Adam optimizer to minimize the $\ell_2$ loss between voxel values and INRs approximations. For a fair comparison, the same training strategy is adopted for FR, BN and IGA. Detailed settings can be found in our Appendix H.

In Table 2, we report the average intersection over union (IOU) and Chamfer Distance metrics of five objects for reference. Under our training settings, baseline models optimized by vanilla gradient have converged to a favorable optimum, significantly outperforming prior works (Saragadam et al., 2023; Cai et al., 2024; Shi et al., 2024a). Nevertheless, our IGA method enables models to explore superior optima by purposefully improving the spectral bias, thereby leading to improvements in representation accuracy such as sharper edges in Fig. 5. More results are in our Appendix H.

## 5.3. Learning 5D neural radiance fields

Learning neural radiance fields for novel view synthesis, i.e., NeRF, is the main application of INRs (Mildenhall et al., 2021; Saragadam et al., 2023; Xie et al., 2023; Shi et al., 2024a). The main process of NeRF is to build a neural radiance field from a 5D coordinate space to RGB space.

Specifically, given a ray $i$ from the camera into the scene, the MLP takes the 5D coordinates (3D spatial locations and 2D view directions) of points along the ray as inputs and outputs the corresponding color **c** and volume density $\sigma$. Then **c** and $\sigma$ are combined using numerical volume rendering to obtain the final pixel color of the ray $i$. Following our

sampling strategy, We place the 2D view directions last during flattening and set $p$ as the number of sampling points per ray. That is, each ray is treated as a group. For each group, we sample the point with the maximum integral weight. We apply our method to the original NeRF (Mildenhall et al., 2021). The "NeRF-PyTorch" codebase (Yen-Chen, 2020) is used, and we follow its default settings for all methods. More details can be found in Appendix I.

*Table 3.* Average metrics of 5D neural radiance fields by different methods. Per-scene results are provided in Table. 14.

| Metrics | NeRF | NeRF+FR | NeRF+BN† | **NeRF+IGA** |
|---|---|---|---|---|
| PSNR ↑ | 31.23 | 31.35 | 31.37 | **31.47** |
| SSIM ↑ | 0.953 | 0.954 | 0.956 | **0.955** |
| LPIPS ↓ | $0.029^{Alex}$ | $0.028^{Alex}$ | 0.050 | **$0.027^{Alex}$** |

† Use values reported in the paper (Cai et al., 2024).

Table 3 lists average metrics of four methods on the downscaled Blender dataset (Mildenhall et al., 2021). Our IGA method achieves the best results among these methods. The average improvement of our method is up to $0.24$dB, which is nearly twice that of FR and BN. Visualization results in our Appendix I show that our IGA enables NeRF to capture more accurate and high-frequency reconstruction results.

### 5.4. Time and memory analysis

In this subsection, we analyze the training time and memory cost of our IGA in the aforementioned experiments. For both vanilla gradient descent and IGA, we present the average per-iteration training time and memory cost over all models within each experiment, as summarized in Table 4. We only report the results of adjustments using the complete eNTK matrix (+Full eNTK) for 1D simple function approximation, due to its excessive memory cost (exceeding $80$GiB) and prohibitive training time in other INR tasks.

*Table 4.* Average per-iteration training time and memory cost by different methods. "Vanilla" denotes standard training using conventional gradients; "+Full eNTK" denotes training with gradient adjustment based on the complete eNTK matrix $\boldsymbol{K}$; "+IGA" denotes training with inductive gradient adjustment. The abbreviations "1D", "2D", "3D", and "5D" are used to represent the experiments on simple functions, 2D images, 3D shapes, and neural radiance fields, respectively.

| INR | Training time (s) | | | Memory cost (MB) | | |
|---|---|---|---|---|---|---|
| tasks | Vanilla | +Full eNTK | **+IGA** | Vanilla | +Full eNTK | **+IGA** |
| 1D | 0.028 | 0.185 | **0.044** | 965 | 5861 | **1574** |
| 2D | 0.061 | – | **0.088** | 3703 | – | **5174** |
| 3D | 0.118 | – | **0.169** | 2291 | – | **2468** |
| 5D | 0.159 | – | **0.286** | 9451 | – | **11033** |

Overall, IGA averages $1.56\times$ the training time and $1.34\times$ the memory of the baseline but achieves greater improvements (on average, $2.0\times$ those of prior FR and BN). Compared to "+Full eNTK", IGA reduces the training time and memory consumption by at least $4\times$.

## 6. Conclusion

In this paper, we propose an effective gradient adjustment strategy to purposefully improve the spectral bias of multilayer perceptrons (MLPs) for better implicit neural representations (INRs). We delve into the linear dynamics model of MLPs and theoretically identify that the empirical Neural Tangent Kernel (eNTK) matrix connects spectral bias with the linear dynamics of MLPs. Based on eNTK matrix, we propose our inductive gradient adjustment method via inductive generalization of gradient adjustments from sampled data. Both theoretical and empirical analysis are conducted to validate impacts of our method on spectral bias. Further, we validate our method on various real-world vision applications of INRs. Our method can effectively tailor improvements on spectral bias and lead to better representation for common INRs network architectures. We hope this study inspires further work on neural network training dynamics to improve application performance.

## Impact Statement

This paper presents work whose goal is to advance the field of Machine Learning. There are many potential societal consequences of our work, none which we feel must be specifically highlighted here.

## Acknowledgments

This work was supported by National Natural Science Foundation of China (No. 62476051) and Sichuan Natural Science Foundation (No. 2024NSFTD0041).

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

# Appendix

This appendix is organized as follows. In Sec. A, we provide the formal version and detailed proof of Theorem 3.1 and 3.2 in the main paper. Then, more empirical evidence from Sec. 4 are provided in Sec. B to support theoretical results in Sec. 3.3. In Sec. C, the visual illustration of our sampling strategy is provided. In Sec. D, a further discussion about our inductive gradient adjustment (IGA) with Adam optimizer referred in Sec. 3.4 is presented. Then, we add experiments of IGA on recent state-of-the-art activations in Sec. E. In Sec. F, the ablation study on sampling strategy is provided. Lastly, we give detailed training settings and per-scene results of our three vision applications in Sec. G, H, I, respectively.

## A. Theoretical Analysis

In this section, we first introduce the detailed framework for theoretical analysis. Then, we provide the formal statements and proofs of Theorem 3.1 and 3.2 of Sec. 3.3.

### A.1. Analysis framework

For the sake of simplicity and to focus on the core aspects of the problem, we follow the previous works (Lee et al., 2019; Geifman et al., 2023; Arora et al., 2019), and perform a theoretical analysis on the basis of the following framework: assuming that the training set $\{\boldsymbol{x}_i, y_i\}_{i=1}^N$ is contained in one compact set, a two-layer network $f(\boldsymbol{x}; \Theta)$ with $m$ neurons is formalized as follows to fit these data:

$$f(\boldsymbol{x}; \Theta) = \frac{1}{\sqrt{m}} \sum_{r=1}^m a_r \sigma(\boldsymbol{w}_r^\top \boldsymbol{x} + b_r), \tag{9}$$

where the activation function $\sigma$ satisfies that $|\sigma(0)|, \|\sigma'\|_\infty, sup_{x \neq x'}|\sigma'(x) - \sigma'(x')|/|x - x'| < \infty$. The parameters of $f(\boldsymbol{x}; \Theta)$ are randomly initialized with $\mathcal{N}(0, \frac{c_\sigma}{m})$, except for biases initialized with $\mathcal{N}(0, c_\sigma)$, where $c_\sigma = 1/\mathbb{E}_{z \sim \mathcal{N}(0,1)}[\sigma(z)^2]$. The loss function is $\frac{1}{2} \sum_{i=1}^N (f(\boldsymbol{x}_i, \Theta_t) - y_i)^2$. Then, the NTK matrix $\boldsymbol{K}$ and the empirical NTK matrix $\tilde{\boldsymbol{K}}$ is defined as follows:

$$\boldsymbol{K} = \mathbb{E}_{\Theta_0}[\nabla_{\Theta_0} f(\boldsymbol{X}; \Theta_0)^\top \nabla_{\Theta_0} f(\boldsymbol{X}; \Theta_0)] \tag{10}$$

$$\tilde{\boldsymbol{K}} = \nabla_{\Theta_t} f(\boldsymbol{X}; \Theta_t)^\top \nabla_{\Theta_t} f(\boldsymbol{X}; \Theta_t), \tag{11}$$

where $\nabla_\Theta f(\boldsymbol{X}; \Theta) \in \mathbb{R}^{q \times N}$; $q$ is the number of parameters. We assume that $\boldsymbol{K}, \tilde{\boldsymbol{K}}$ are full rank, indicating that they are positive definite. This assumption generally holds due to the complexity of neural networks. Then we have the following standard orthogonal spectral decomposition: $\boldsymbol{K} = \sum_{i=1}^N \lambda_i \boldsymbol{v}_i \boldsymbol{v}_i^\top$ and $\tilde{\boldsymbol{K}} = \sum_{i=1}^N \tilde{\lambda}_i \tilde{\boldsymbol{v}}_i \tilde{\boldsymbol{v}}_i^\top$, which eigenvalues are indexed in descending order of magnitude. We construct the corresponding transformation matrices as $\boldsymbol{S} = \sum_{i=1}^N \frac{g_i(\lambda_i)}{\lambda_i} \boldsymbol{v}_i \boldsymbol{v}_i^\top$ and $\tilde{\boldsymbol{S}} = \sum_{i=1}^N \frac{g_i(\tilde{\lambda}_i)}{\tilde{\lambda}_i} \tilde{\boldsymbol{v}}_i \tilde{\boldsymbol{v}}_i^\top$ through a set of Lipschitz continuous functions $\{g_i(\lambda)\}_{i=1}^N$. From Jacot et al. (2018); Du et al. (2018); Arora et al. (2019); Geifman et al. (2023), we have that:

$$\lim_{m \to \infty} \|\boldsymbol{K} - \tilde{\boldsymbol{K}}\|_F = 0. \tag{12}$$

### A.2. Basic Lemmas

In this section, we show some basic lemmas. Lemma A.1 describes the local properties of $f(\boldsymbol{x}, \Theta)$. Based on Lemma A.1, we derive Lemma A.2, A.3 and A.4, which characterize training dynamics of $f(\boldsymbol{x}, \Theta)$ under different adjusted gradients.

**Lemma A.1.** *(Modified from Lemma A.1 of (Geifman et al., 2023)) For bounded matrices, i.e., $\boldsymbol{I}, \boldsymbol{S}, \tilde{\boldsymbol{S}}$, there is a $\kappa > 0$ such that for every $C > 0$, with high probability over random initialization the following holds: $\forall \Theta, \tilde{\Theta} \in B(\Theta_0, Cm^{-\frac{1}{2}})$ at time step $t$:*

$$\|(\nabla_\Theta f(\boldsymbol{X}) - \nabla_{\tilde{\Theta}} f(\boldsymbol{X})) \boldsymbol{A}\|_F \leq \frac{\kappa}{\sqrt{m}} \|\Theta - \tilde{\Theta}\|_F \tag{13}$$

$$\|\nabla_\Theta f(\boldsymbol{X}) \boldsymbol{A}\|_F \leq \frac{\kappa}{\sqrt{m}}, \tag{14}$$

*where $\boldsymbol{A}$ can be $\boldsymbol{I}, \boldsymbol{S}, \tilde{\boldsymbol{S}}$.*

As discussed in Lee et al. (2019); Geifman et al. (2023), the core of Lemma A.1 is the requirement that the matrix $\boldsymbol{A}$ is bounded. Therefore, Lemma A.1 clearly holds. Then, to better illustrate our core theoretical analysis, we present the following Lemma A.2, A.3 and A.4. Please note that a similar version of Lemma A.2 has been proved in (Geifman et al., 2023). However, it focuses on $\boldsymbol{r}_{t+1}$ induced by only $\boldsymbol{S}$ and the details of the eigenvalues between $\boldsymbol{r}_{t+1}$ and $\boldsymbol{r}_t$ are omitted. Therefore, our Lemma A.2, A.3 and A.4 characterize the dynamics in one iteration from $\boldsymbol{r}_t$ to $\boldsymbol{r}_{t+1}$ and eigenvalues induced by NTK-based and eNTK-based adjustments and vanilla gradient, respectively.

**Lemma A.2.** *The parameters are updated by:* $\Theta_{t+1} = \Theta_t - \eta \nabla_{\Theta_t} f(\boldsymbol{X}, \Theta_t) \boldsymbol{S} \boldsymbol{r}_t$ *with* $\eta < (min(g_i(\lambda)) + max(g_i(\lambda)))^{-1}$. *For* $\epsilon > 0$, *there always exists* $M > 0$, *when* $m > M$, *such that with high probability over the random initialization, we have that :*

$$\|\boldsymbol{r}_{t+1}\|_2^2 = \sum_{i=1}^{N}(1 - \eta g_i(\lambda_i))^2(\boldsymbol{v}_i^\top \boldsymbol{r}_t)^2 \pm \xi(t), \tag{15}$$

*where* $\boldsymbol{r}_t = (f(\boldsymbol{x}_i, \Theta_t) - y_i)_{i=1}^N$ *and* $|\xi(t)| < \epsilon$.

*Proof.* By the mean value theorem with respect to parameters $\Theta$, we can have that:

$$\boldsymbol{r}_{t+1} = \boldsymbol{r}_{t+1} - \boldsymbol{r}_t + \boldsymbol{r}_t = \nabla_{\Theta_t'} f(\boldsymbol{X})^\top(\Theta_{t+1} - \Theta_t) + \boldsymbol{r}_t = \nabla_{\Theta_t'} f(\boldsymbol{X})^\top(-\eta \nabla_{\Theta_t} f(\boldsymbol{X}) \boldsymbol{S} \boldsymbol{r}_t) + \boldsymbol{r}_t$$
$$= (\boldsymbol{I} - \eta \boldsymbol{K} \boldsymbol{S})\boldsymbol{r}_t + \underbrace{\eta(\boldsymbol{K} - \tilde{\boldsymbol{K}})\boldsymbol{S}\boldsymbol{r}_t}_{A} + \underbrace{\eta(\nabla_{\Theta_t} f(\boldsymbol{X}) - \nabla_{\Theta_t'} f(\boldsymbol{X}))^\top \nabla_{\Theta_t} f(\boldsymbol{X}) \boldsymbol{S} \boldsymbol{r}_t}_{B},$$

where $\Theta_t'$ lies on the line segment $\overline{\Theta_t \Theta_{t+1}}$. We define that $\boldsymbol{\xi}'(t) = A + B$. For $\|A\|$, we have that:

$$\|A\| \leq \eta\|(\boldsymbol{K} - \tilde{\boldsymbol{K}})\boldsymbol{S}\boldsymbol{r}_t\| \leq \eta\|\boldsymbol{K} - \tilde{\boldsymbol{K}}\|_F\|\boldsymbol{S}\|_F\|\boldsymbol{r}_t\|_2$$
$$\leq^{(1)} \eta L R_0\|\boldsymbol{K} - \tilde{\boldsymbol{K}}\|_F \leq^{(2)} \frac{\epsilon}{2},$$

where (1) follows Geifman et al. (2023) that $\boldsymbol{S}$ and $\boldsymbol{r}_t$ are bounded by constants $L$ and $R_0$, respectively; (2) follows the equation 12.

For $\|B\|$, we have that:

$$\|B\| \leq \eta\|\nabla_{\Theta_t} f(\boldsymbol{X}, \Theta_t) - \nabla_{\Theta_t'} f(\boldsymbol{X}, \Theta_t')\|_F\|\nabla_{\Theta_t} f(\boldsymbol{X})\boldsymbol{S}\|_F\|\boldsymbol{r}_t\|_2 \leq \frac{\eta \kappa^2 R_0}{m}\|\Theta_t - \Theta_t'\|_2$$
$$\leq \frac{\eta \kappa^2 R_0}{m}\|\Theta_t - \Theta_{t+1}\|_2 \leq \frac{\eta \kappa^2 R_0}{m}\|\eta \nabla_{\Theta_t} f(\boldsymbol{X}, \Theta_t)\boldsymbol{S}\boldsymbol{r}_t\| \leq^{(1)} \frac{\eta^2 \kappa^3 R_0^2}{m^{3/2}},$$

where (1) follows the Lemma A.1. Therefore, we have that $\boldsymbol{r}_{t+1} = \sum_{i=1}^{N}(1 - \eta g_i(\lambda_i))(\boldsymbol{v}_i^\top \boldsymbol{r}_t)\boldsymbol{v}_i \pm \boldsymbol{\xi}'(t)$; there always exits $M_1 > 0$, such that $m > M_1$, $\|\boldsymbol{\xi}'(t)\| \leq \|A\| + \|B\| < \epsilon$. Further, we have that:

$$\|\boldsymbol{r}_{t+1}\|_2^2 = \sum_{i=1}^{N}(1 - \eta g_i(\lambda_i))^2(\boldsymbol{v}_i^\top \boldsymbol{r}_t)^2 + \|\boldsymbol{\xi}'(t)\|_2^2 \pm 2\boldsymbol{\xi}'(t)^\top \sum_{i=1}^{N}(1 - \eta g_i(\lambda_i))(\boldsymbol{v}_i^\top \boldsymbol{r}_t)\boldsymbol{v}_i$$
$$= \sum_{i=1}^{N}(1 - \eta g_i(\lambda_i))^2(\boldsymbol{v}_i^\top \boldsymbol{r}_t)^2 + \underbrace{\|\boldsymbol{\xi}'(t)\|_2^2 \pm 2\boldsymbol{\xi}'(t)^\top(\boldsymbol{r}_t \pm \boldsymbol{\xi}'(t))}_{\xi(t)}.$$

For $\xi(t)$, as $\boldsymbol{r}_{t+1}$ is bounded by $R_0$, there always exists $M > M_1$, such that:

$$|\xi(t)| \leq 3\|\boldsymbol{\xi}'(t)\|_2^2 + 2R_0\|\boldsymbol{\xi}'(t)\|_2^2 < \epsilon.$$

$\square$

**Lemma A.3.** *The parameters are updated by:* $\Theta_{t+1} = \Theta_t - \eta \nabla_{\Theta_t} f(\boldsymbol{X}, \Theta_t) \tilde{\boldsymbol{S}} \boldsymbol{r}_t$ *with* $\eta = (max(g_i(\tilde{\lambda})) + min(g_i(\tilde{\lambda})))^{-1}$. *For* $\epsilon > 0$, *there always exists* $M > 0$, *when* $m > M$, *such that with high probability over the random initialization, we have that :*

$$\|\tilde{\boldsymbol{r}}_{t+1}\|_2^2 = \sum_{i=1}^{N}(1 - \eta g_i(\tilde{\lambda}_i))^2(\tilde{\boldsymbol{v}}_i^\top \boldsymbol{r}_t)^2 \pm \tilde{\xi}(t), \tag{16}$$

*where* $\boldsymbol{r}_t = (f(\boldsymbol{x}_i, \Theta_t) - y_i)_{i=1}^N$ *and* $|\tilde{\xi}(t)| < \epsilon$.

*Proof.* By the mean value theorem with respect to parameters $\Theta$, we can have that:

$$\boldsymbol{r}_{t+1} = \boldsymbol{r}_{t+1} - \boldsymbol{r}_t + \boldsymbol{r}_t = \nabla_{\Theta'_t} f(\boldsymbol{X}, \Theta'_t)^\top (\Theta_{t+1} - \Theta_t) + \boldsymbol{r}_t = \nabla_{\Theta'_t} f(\boldsymbol{X}, \Theta'_t)^\top (-\eta \nabla_{\Theta_t} f(\boldsymbol{X}, \Theta_t) \tilde{\boldsymbol{S}} \boldsymbol{r}_t) + \boldsymbol{r}_t$$

$$= (\boldsymbol{I} - \eta \tilde{\boldsymbol{K}} \tilde{\boldsymbol{S}}) \boldsymbol{r}_t + \underbrace{\eta (\nabla_{\Theta_t} f(\boldsymbol{X}, \Theta_t) - \nabla_{\Theta'_t} f(\boldsymbol{X}, \Theta'_t))^\top \nabla_{\Theta_t} f(\boldsymbol{X}) \tilde{\boldsymbol{S}} \boldsymbol{r}_t}_{\tilde{\boldsymbol{\xi}}'(t)}.$$

For $\tilde{\boldsymbol{\xi}}'(t)$, we follow the same technique in the proof of A.2. Then we can have that:

$$\|\tilde{\boldsymbol{\xi}}'(t)\| \leq \eta \|\nabla_{\Theta_t} f(\boldsymbol{X}) - \nabla_{\Theta'_t} f(\boldsymbol{X})\|_F \|\nabla_{\Theta_t} f(\boldsymbol{X}) \boldsymbol{S}^t\|_F \|\boldsymbol{r}_t\|_2 \leq \eta \frac{\kappa^2}{m} \|\Theta_t - \Theta'_t\|_2$$

$$\leq \frac{\eta \kappa^2 R_0}{m} \|\Theta_t - \Theta_{t+1}\|_2 \leq \frac{\eta^2 \kappa^2 R_0^2}{m^{3/2}}.$$

Therefore, we have that $\boldsymbol{r}_{t+1} = \sum_{i=1}^N (1 - \eta g_i(\tilde{\lambda}_i))(\tilde{\boldsymbol{v}}_i^\top \boldsymbol{r}_t)\tilde{\boldsymbol{v}}_i \pm \tilde{\boldsymbol{\xi}}'(t)$; there always exits $M_1 > 0$, such that $m > M_2$, $\|\tilde{\boldsymbol{\xi}}'(t)\| < \epsilon$. Further, we have that:

$$\|\boldsymbol{r}_{t+1}\|_2^2 = \sum_{i=1}^N (1 - \eta g_i(\tilde{\lambda}_i))^2 (\tilde{\boldsymbol{v}}_i^\top \boldsymbol{r}_t)^2 + \|\tilde{\boldsymbol{\xi}}'(t)\|_2^2 \pm 2\tilde{\boldsymbol{\xi}}'(t)^\top \sum_{i=1}^N (1 - \eta g_i(\tilde{\lambda}_i))(\tilde{\boldsymbol{v}}_i^\top \boldsymbol{r}_t)\tilde{\boldsymbol{v}}_i$$

$$= \sum_{i=1}^N (1 - \eta g_i(\tilde{\lambda}_i))^2 (\tilde{\boldsymbol{v}}_i^\top \boldsymbol{r}_t)^2 + \underbrace{\|\tilde{\boldsymbol{\xi}}'(t)\|_2^2 \pm 2\tilde{\boldsymbol{\xi}}'(t)^\top (\boldsymbol{r}_t \pm \tilde{\boldsymbol{\xi}}'(t))}_{\tilde{\xi}(t)}.$$

For $\xi(t)$, as $\boldsymbol{r}_{t+1}$ is bounded by $R_0$, there always exists $M > M_2$, such that:

$$|\tilde{\xi}(t)| \leq 3\|\tilde{\boldsymbol{\xi}}'(t)\|_2^2 + 2R_0\|\tilde{\boldsymbol{\xi}}'(t)\|_2^2 < \epsilon.$$

$\square$

**Lemma A.4.** *The parameters are updated by vanilla gradient descent. For $\epsilon > 0$, there always exists $M > 0$, when $m > M$, such that with high probability over the random initialization, we have that:*

$$f(\boldsymbol{x}_i, \Theta_{t+1}) - f(\boldsymbol{x}_i, \Theta_t) = -\eta \nabla_{\Theta_t} f(\boldsymbol{x}_i, \Theta_t)^\top \sum_{j=1}^N \nabla_{\Theta_t} f(\boldsymbol{x}_j, \Theta_t)(f(\boldsymbol{x}_j, \Theta_t) - y_j) \pm \xi_i'(t), \tag{17}$$

*where $|\xi_i'(t)| < \epsilon$.*

*Proof.* By the mean value theorem with respect to parameters $\Theta$, we can have that:

$$f(\boldsymbol{x}_i, \Theta_{t+1}) = \nabla_{\Theta'_t} f(\boldsymbol{x}_i, \Theta'_t)^\top (\Theta_{t+1} - \Theta_t) + f(\boldsymbol{x}_i, \Theta_t)$$

$$= \nabla_{\Theta'_t} f(\boldsymbol{x}_i, \Theta'_t)^\top [-\eta \sum_{j=1}^N \nabla_{\Theta_t} f(\boldsymbol{x}_i, \Theta_t)(f(\boldsymbol{x}_j, \Theta_t) - y_j)] + f(\boldsymbol{x}_i, \Theta_t)$$

$$= -\eta \nabla_{\Theta_t} f(\boldsymbol{x}_i, \Theta_t)^\top \sum_{j=1}^N \nabla_{\Theta_t} f(\boldsymbol{x}_j, \Theta_t)(f(\boldsymbol{x}_j, \Theta_t) - y_j) + f(\boldsymbol{x}_i, \Theta_t) + \xi_i'(t).$$

For $\xi_i'(t)$, we have that:

$$|\xi_i'(t)| = |(\nabla_{\Theta'_t} f(\boldsymbol{x}_i, \Theta'_t) - \nabla_{\Theta_t} f(\boldsymbol{x}_i, \Theta_t))^\top \sum_{j=1}^N \nabla_{\Theta_t} f(\boldsymbol{x}_j, \Theta_t)(f(\boldsymbol{x}_j, \Theta_t) - y_j)|$$

$$\leq \|\nabla_{\Theta'_t} f(\boldsymbol{x}_i, \Theta'_t) - \nabla_{\Theta_t} f(\boldsymbol{x}_i, \Theta_t)\|_2 \|\sum_{j=1}^N \nabla_{\Theta_t} f(\boldsymbol{x}_j, \Theta_t)(f(\boldsymbol{x}_j, \Theta_t) - y_j)\|_2$$

$$\leq \|\nabla_{\Theta'_t} f(\boldsymbol{x}_i, \Theta'_t) - \nabla_{\Theta_t} f(\boldsymbol{x}_i, \Theta_t)\|_2 \|\nabla_{\Theta_t} f(\boldsymbol{X}, \Theta_t)\|_2 \|\boldsymbol{r}_t\|_2$$

$$\leq^{(1)} \frac{\kappa^2 R_0}{m} \|\Theta_{t+1} - \Theta_t\|_2 \leq^{(2)} \frac{\eta \kappa^3 R_0^2}{m^{3/2}},$$

where $(1), (2)$ follow the Lemma A.1 and $r_t$ is bounded by $R_0$; Therefore, there exists $M > 0$, such that $m > M$, we have that $|\xi_i'(t)| < \epsilon$. $\qquad\square$

Lemma A.2 and lemma A.3 show the training dynamics of the residual $r_t$ by NTK-based and eNTK-based gradient adjustments. Equation 15 shows that $\{g_i(\lambda_i)\}_{i=1}^N$ control the decay rates of different frequency components of the residual $r_t$ as eigenvectors of $K$ are the spherical harmonics (Ronen et al., 2019). Lemma A.4 show the training dynamics of residual at data point $x_i$ by vanilla gradient descent. Despite eNTK-based and NTK-based gradient adjustments have the similar form, they differ in the eigenvector directions $\tilde{v}_i$, which leads to errors. We give a theoretical analysis to this error in Theorem 3.1.

## A.3. Proof of Theorem 3.1

**Theorem A.5.** *(formal version of Theorem 3.1 from the paper) The following standard orthogonal spectral decomposition exists:* $K = \sum_{i=1}^N \lambda_i v_i v_i^\top$ *and* $\tilde{K} = \sum_{i=1}^N \tilde{\lambda}_i \tilde{v}_i \tilde{v}_i^\top$, *satisfying that* $v_i^\top \tilde{v}_i > 0$ *for* $i \in [N]$. *We denoted as* $\max_i\{\min\{\lambda_i - \lambda_{i+1}, \lambda_{i+1} - \lambda_{i+2}\}\} = G$. *Let* $\{g_i(x)\}_{i=1}^N$ *be a set of Lipschitz continuous functions, with the Supremum of their Lipschitz constants denoted by* $k$, *with* $\eta < \min\{(\max(g_i(\lambda)) + \min(g_i(\lambda)))^{-1}, (\max(g_i(\tilde{\lambda})) + \min(g_i(\tilde{\lambda})))^{-1}\}$, *for* $\epsilon > 0$, *there always exists* $M > 0$, *such that* $m > M$, *for* $i \in [N]$, *we have that* $|g_i(\lambda_i) - g_i(\tilde{\lambda}_i)| < \epsilon_1$, $\|v_i - \tilde{v}_i\| < \epsilon_2$; *furthermore, we have that:*

$$|(1 - \eta g_i(\tilde{\lambda}_i))^2 (\tilde{v}_i^\top r_t)^2 - (1 - \eta g_i(\lambda_i))^2 (v_i^\top r_t)^2| < \epsilon_3, \|r_{t+1} - \tilde{r}_{t+1}\| < \epsilon_4,$$

*where* $\epsilon_1 = k\epsilon$; $\epsilon_2 = (2^{3/2}\epsilon)/G$; $\epsilon_3 = \frac{8R_0}{G^2}(k\eta\epsilon^3 + (v+1)\epsilon^2 + (v^2 + Gv)\epsilon)$; $\epsilon_4 = \epsilon N((\frac{16vR_0}{G} + \eta kv^2 R_0) + \epsilon^2) + 2\epsilon$; $v, k, R_0$ *are constants.*

*Proof.* We define that the perturbation of $K$ as $E = K - \tilde{K}$. As previously discussed, we have that $\lim_{m\to\infty}\|K - \tilde{K}\|_F = 0$. Therefore, for any $\epsilon > 0$ there always exists $M_0$, such that $m > M_0$, we have that $\|E\|_F < \epsilon$. As $K, \tilde{K}$ are real symmetric matrices, $E$ is still a real symmetric matrix. This indicates that:

$$\|E\|_2 = \sqrt{(\lambda_1')^2} \le \sqrt{\sum_{i=1}^N (\lambda_i')^2} = \sqrt{trace(E^\top E)} = \|E\|_F,$$

where $\lambda_i'$ is the largest eigenvalue of $E$.

By Wely's inequality, when $m > M_0$, for $i = 1, \ldots, N$, we have that $|\lambda_i - \tilde{\lambda}_i| \le \|E\|_2 < \epsilon$. As $g_i(\lambda)$ is a Lipschitz continuous function with the Lipschitz constant $k$, for $i \in [N]$, we have that:

$$|g_i(\lambda_i) - g_i(\tilde{\lambda}_i)| \le k|\lambda_i - \tilde{\lambda}_i| < k\epsilon.$$

Therefore, for any $0 < \epsilon$, there always exists $M_0$, such that $|g(\lambda_i) - g(\tilde{\lambda}_i)| < k\epsilon$. By the Corollary 3. of Yu et al. (2015) that a variant of Davis-Kahan theorem, we have that:

$$\|\tilde{v}_i - v_i\| \le \frac{2^{3/2}\|E\|_2}{G} < \frac{2^{3/2}\epsilon}{G}, i \in [N].$$

Then, we have that:

$$\begin{aligned}
&|(1 - \eta g_i(\lambda_i))^2 (v_i^\top r_t)^2 - (1 - \eta g_i(\tilde{\lambda}_i))^2 (\tilde{v}_i^\top r_t)^2| \\
=&|(1 - \eta g_i(\lambda_i))^2 (v_i^\top r_t)^2 - (1 - \eta g_i(\lambda_i))^2 (\tilde{v}_i^\top r_t)^2 + (1 - \eta g_i(\lambda_i))^2 (\tilde{v}_i^\top r_t)^2 - (1 - \eta g_i(\tilde{\lambda}_i))^2 (\tilde{v}_i^\top r_t)^2| \\
<& \underbrace{|(1 - \eta g_i(\lambda_i))(v_i^\top r_t + \tilde{v}_i^\top r_t)(v_i^\top r_t - \tilde{v}_i^\top r_t)|}_{A} + \underbrace{|\eta(g_i(\tilde{\lambda}_i) - g_i(\lambda_i))(\tilde{v}_i^\top r_t)^2|}_{B}.
\end{aligned}$$

For $A$, we have that:

$$\begin{aligned}
A &< |1 - \eta g_i(\lambda_i)| \cdot \|v_i^\top + \tilde{v}_i^\top\|_2 \|v_i^\top - \tilde{v}_i^\top\|_2 \|r_t\|_2^2 \\
&< |1 - \eta g_i(\lambda_i)| \cdot (2\|v_i^\top\| + \|\tilde{v}_i^\top - v_i\|_2)\|\tilde{v}_i^\top - v_i\|_2 \|r_t\|_2^2 < \epsilon\frac{8R_0^2(G\|v_i^\top\| + \epsilon)}{G^2}.
\end{aligned}$$

As $S$ is fixed, we have that a constant $v$ to bound $\|v_i\|$, for $i \in [N]$. Therefore, for $\epsilon > 0$, there always exists $M$, such that $m > M_0$, we have that $A < \frac{8R_0^2(\epsilon Gv + \epsilon^2)}{G^2}$.

For B, we have that:

$$B < \eta|g_i(\tilde{\lambda}_i) - g_i(\lambda_i)|(\tilde{v}_i^\top r_t)^2 < \epsilon k \eta (\|\tilde{v}_i^\top - v_i^\top\|\|r_t\| + \|v_i\|\|r_t\|)^2 < \epsilon k \eta R_0 (\frac{2^{3/2}}{G}\epsilon + v)^2.$$

Further, we have that:

$$|(1 - \eta g_i(\lambda_i))^2(v_i^\top r_t)^2 - (1 - \eta g_i(\tilde{\lambda}_i))^2(\tilde{v}_i^\top r_t)^2| < \frac{8R_0}{G^2}(k\eta\epsilon^3 + (v+1)\epsilon^2 + (v^2 + Gv)\epsilon).$$

Following these analytical approaches, it is trivial to have the following result:

$$|(1 - \eta g(\lambda_i))(v_i^\top r_t)v_i - (1 - \eta g(\tilde{\lambda}_i))(\tilde{v}_i^\top r_t)\tilde{v}_i| < \epsilon(\frac{16vR_0}{G} + \eta k v^2 R_0) + \epsilon^2.$$

Combing this result with the Lemma A.2 and A.3, there always exists $M > max\{M_0, M_1, M_2\}$, we have that:

$$\|r_{t+1} - \tilde{r}_{t+1}\| = \|\sum_{i=1}^N [(1 - \eta g(\lambda_i))(v_i^\top r_t)v_i - (1 - \eta g(\tilde{\lambda}_i))(\tilde{v}_i^\top r_t)\tilde{v}_i] \pm \xi'(t) \pm \tilde{\xi}'(t)\|$$

$$< \epsilon N((\frac{16vR_0}{G} + \eta k v^2 R_0) + \epsilon^2) + \|\xi'(t)\| + \|\tilde{\xi}'(t)\|$$

$$< \epsilon N((\frac{16vR_0}{G} + \eta k v^2 R_0) + \epsilon^2) + 2\epsilon.$$

$\square$

## A.4. Proof of Theorem 3.2

**Theorem A.6.** *(formal version of Theorem 3.2 from the paper)* $X = \{x_i\}_{i=1}^N$ *is partitioned into $n$ groups, where each group is $X_j = \{x_1^j, \ldots, x_p^j\}$ and $N = np$. For each group, one data point is sampled denoted as $x^j$ and these $n$ data points form $X_e = \{x^j\}_{j=1}^n, n \ll N$. There exists $\epsilon > 0$, for $j = 1, \ldots, n$ and any $1 \le i_1, i_2 \le p$, such that $|(f(x_{i_1}^j, \Theta_t) - y_{i_1}^j) - (f(x_{i_2}^j, \Theta_t) - y_{i_2}^j)|, \|\nabla_{\Theta_t} f(x_{i_1}^j) - \nabla_{\Theta_t} f(x_{i_2}^j)\| < \epsilon$. Then, for any $x_i$, assumed that $x_i$ belongs to $X_{j_i}$, such that:*

$$|\Delta f(x_i, \Theta_t) - \nabla_{\Theta_t} f(x^{j_i}, \Theta_t)^\top [p \sum_{j=1}^n \nabla_{\Theta_t} f(x^j, \Theta_t)(f(x^j) - y^j)]| < \epsilon(\frac{(\kappa + n^{3/2})R_0}{\sqrt{m}}) + \frac{\eta\kappa^3 R_0^2}{m^{3/2}},$$

*where $\Delta f(x_i, \Theta_t) = (r_{t+1} - r_t)_i$ denotes the dynamics of $f(x_i, \Theta)$ at time step $t$; $R_0, \kappa$ are constants; $\nabla_{\Theta_t} f(x^{j_i}, \Theta)^\top \nabla_{\Theta_t} f(x^j, \Theta)$ is the entry in $\tilde{K}_e(i, j)$.*

*Proof.* From the equation 17 of lemma A.4, we consider the following error:

$$|\nabla_{\Theta_t} f(x_i, \Theta_t)^\top \sum_{i=1}^N \nabla_{\Theta_t} f(x_i, \Theta_t)(f(x_i, \Theta_t) - y_i) - \nabla_{\Theta_t} f(x^{j_i}, \Theta_t)^\top [p \sum_{j=1}^n \nabla_{\Theta_t} f(x^j, \Theta_t)(f(x^j) - y^j)]|.$$

Following previous analysis technique, this error can be bounded by $A + B$. For $A$, we have that:

$$A = \|(\nabla_{\Theta_t} f(x_i, \Theta_t)^\top - \nabla_{\Theta_t} f(x^{j_i}, \Theta_t)^\top) \sum_{j=1}^N \nabla_{\Theta_t} f(x_j, \Theta_t)(f(x_j, \Theta_t) - y_j)\|$$

$$\le \|\nabla_{\Theta_t} f(x_i, \Theta_t) - \nabla_{\Theta_t} f(x^{j_i}, \Theta_t)\|\|\sum_{j=1}^N \nabla_{\Theta_t} f(x_j, \Theta_t)(f(x_j, \Theta_t) - y_j)\|$$

$$\le \epsilon\|\nabla_{\Theta_t} f(X, \Theta_t)Ir_t\| \le \epsilon\frac{\kappa R_0}{\sqrt{m}}.$$

For $B$, we have that:

$$B = \|\nabla_{\Theta_t} f(\boldsymbol{x}^{j_i}, \Theta_t)^\top (\sum_{j=1}^N \nabla_{\Theta_t} f(\boldsymbol{x}_j, \Theta_t)(f(\boldsymbol{x}_j, \Theta_t) - y_j) - p\sum_{j=1}^n \nabla_{\Theta_t} f(\boldsymbol{x}^j, \Theta_t)(f(\boldsymbol{x}^j) - y^j))\|$$

$$\leq \|\nabla_{\Theta_t} f(\boldsymbol{x}^{j_i}, \Theta_t)^\top\| \cdot \sum_{j=1}^p \sum_{i=1}^n \|\nabla_{\Theta_t} f(\boldsymbol{x}^j, \Theta_t)(f(\boldsymbol{x}^j, \Theta_t) - y^j) - \nabla_{\Theta_t} f(\boldsymbol{x}_i^j, \Theta_t)(f(\boldsymbol{x}_i^j, \Theta_t) - y_i^j)\| \quad (18)$$

$$\leq \|\nabla_{\Theta_t} f(\boldsymbol{x}^{j_i}, \Theta_t)^\top\| \cdot \sum_{j=1}^p \sum_{i=1}^n \epsilon(|f(\boldsymbol{x}^j, \Theta_t) - y^j| + \|\nabla_{\Theta_t} f(\boldsymbol{x}_i^j, \Theta_t)\|)$$

$$\leq \epsilon\|\nabla_{\Theta_t} f(\boldsymbol{x}^{j_i}, \Theta_t)^\top\|(n^{3/2} R_0 + \frac{\kappa}{\sqrt{m}}) \leq \epsilon(\frac{n^{3/2} R_0 \kappa}{\sqrt{m}} + \frac{\kappa^2}{m}).$$

We define that $\epsilon' = A + B$. By the lemma A.4, we have that:

$$f(\boldsymbol{x}_i, \Theta_{t+1}) - f(\boldsymbol{x}_i, \Theta_t) = \nabla_{\Theta_t} f(\boldsymbol{x}^{j_i}, \Theta_t)^\top [p\sum_{j=1}^n \nabla_{\Theta_t} f(\boldsymbol{x}^j, \Theta_t)(f(\boldsymbol{x}^j) - y^j)] \pm \epsilon' \pm \xi_i'(t),$$

where $|\epsilon'| \leq \epsilon(\frac{\kappa R_0}{\sqrt{m}} + \frac{n^{3/2} R_0 \kappa}{\sqrt{m}} + \frac{\kappa^2}{m}) \leq \epsilon(\frac{(\kappa + n^{3/2}) R_0 + \kappa^2}{\sqrt{m}})$; $|\xi_i'(t)| \leq \frac{\eta \kappa^3 R_0^2}{m^{3/2}}$. Based on this, the following vector form can be derived:

$$\boldsymbol{r}_{t+1}^i = p\tilde{\boldsymbol{K}}_e \boldsymbol{r}_t^e \pm \boldsymbol{\epsilon}' \pm \boldsymbol{\xi}'(\boldsymbol{t}), \quad (19)$$

where $\boldsymbol{r}_{t+1}^i = (f(\boldsymbol{x}_i^j, \Theta_t) - y_i^j)_{j=1}^m, \boldsymbol{r}_t^e = (f(\boldsymbol{x}^j, \Theta_t) - y^j)$. $\qquad \square$

Further, from the derived expression equation 18, we observe that for different $\boldsymbol{r}_{t+1}^i$ we can replace $\boldsymbol{r}_t^e$ with $\boldsymbol{r}_t^i$ as the pairwise cancellation of some error terms.

## B. Empirical analysis on simple function approximation

In this section, we first visualize more results of the experiment 1 in our paper. Subsequently, we illustrate impacts of IGA on spectral bias of ReLU and SIREN with varying numbers of balanced eigenvalues under varying group size $p$ from the experiment 2. As can be seen in Fig. 9, the same trend as our observations in our paper, with the width increases, the differences on MSE curves and spectrum between gradient adjustments by $\boldsymbol{S}$, $\tilde{\boldsymbol{S}}$ and $\tilde{\boldsymbol{S}}_e$ gradually diminish when $end = 12, 14$. This is consistent with our Theorem 3.1 and the analysis of our Theorem 3.2. As shown in Fig. 10, 11 and 12, impacts on spectral bias are gradually amplified by increasing the number of balanced eigenvalues of $\tilde{\boldsymbol{S}}_e$, which is also consistent with the observations in our paper. The above empirical results corroborate our theoretical results and show that our IGA method provides a chance to amplify impacts on spectral bias by increasing the number of balanced eigenvalues under varying group size $p$.

## C. Visual illustration for the sampling strategy of IGA

In this section, we aim to provide a visual illustration of the sampling strategy introduced in Sec. 3.4. As introduced in the main text, for 1D and 2D inputs, we sample data with the largest residuals in each non-overlapping interval and patch, respectively. For higher-dimensional settings like $N$ $d$-dimensional input coordinates ($d \geq 3$), represented as a tensor of shape $(n_1, \ldots, n_d, d)$, we flatten the first $d$ dimensions into a 2D tensor $\boldsymbol{X}'$ of shape $(N, d)$, where $N = n_1 \times \cdots \times n_d$ (which is evident), and then group it along the first dimension of $\boldsymbol{X}'$. Then, the input with the largest residuals in each group are selected to form the sampled data $\boldsymbol{X}_e$. We take $d = 3$ as an example to further illustrate our sampling strategy. Considering a 3D "Thai" statue, we first sample data over a $512 \times 512 \times 512$ volume grid with each voxel within the "Thai" volume assigned a 1, and voxel outside the "Thai" volume assigned a 0 following Saragadam et al. (2023); Shi et al. (2024a). The 3D coordinates $\{x_1, \ldots, x_{n_1}\} \times \{y_1, \ldots, y_{n_2}\} \times \{z_1, \ldots, z_{n_3}\}$ of voxels defined by the "Thai" volume are the inputs of INRs. We denote these coordnates as $\boldsymbol{X}$. Generally, $\boldsymbol{X}$ is represented as tensor of shape $(n_1, n_2, n_3, 3)$ and $n_1, n_2, n_3$ depends on the resolution of the aforementioned volume grid and the object volume. As demonstrated in Fig. 6,

we visualize $X$ as a cube with dimensions $n_1 \times n_2 \times n_3$, composed of $N$ cubical elements. Each cubical element contains its corresponding coordinate. For example, if we define the $n_1$-dimension as the $x$-axis, the $n_2$-dimension as the $y$-axis, and the $n_3$-dimension as the $z$-axis (with positive directions indicated in Fig. 6), the red-marked cubical element contains the corresponding coordinate $(x_{n_1}, y_1, z_{n_3})$. Then, we use the flatten operation to unfold the cube sequentially along the dimensions, obtaining $X'$. Note that the flatten operation preserves adjacency. Therefore, we directly group $X'$ and sample one element from each group to form $X_e$, marked green in Fig. 6.

This sampling strategy is simple yet effective. The adjacency during grouping helps obtain a smaller consistency bound $\epsilon$, thereby reducing the estimate error. Meanwhile, the flatten operation maintains adjacency while reducing computational overhead. Our extensive experimental results validate the effectiveness of this strategy. Note that there might be more sophisticated strategies via clustering or segmentation, which diverge from the primary focus of our work. We leave this for future research. The ablation study about hyperparameters $n$ and $p$ of the strategy are provided in Sec. F.

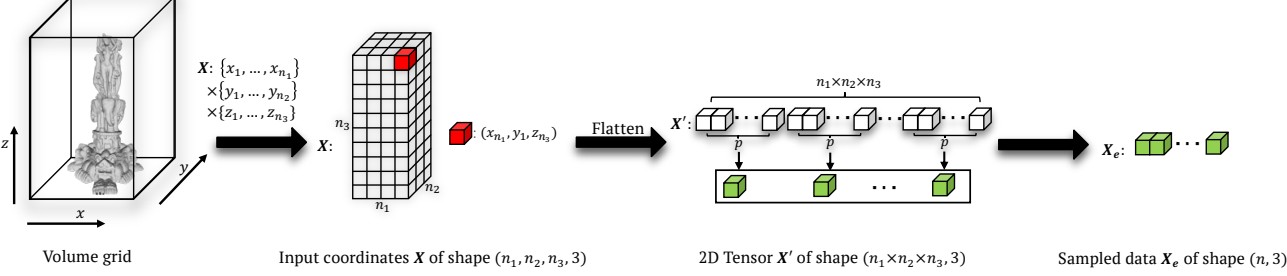

*Figure 6.* Visual illustration for the sampling strategy of IGA. The detailed description of the above diagram is in Sec. C.

## D. Discussion about IGA in Adam

In the current field of deep learning, Adam (Kingma, 2014) as a variant of vanilla gradient descent has been proven to be one of the most effective methods empirically (Deng et al., 2009; He et al., 2016; Vaswani, 2017) and is widely adopted by current INR models (Sitzmann et al., 2020; Mildenhall et al., 2021; Saragadam et al., 2023; Zhu et al., 2024; Shi et al., 2024a). However, due to the introduction of adaptive learning rates and momentum, the characterization of training dynamics like Eq. 17 induced by Adam becomes exceptionally challenging and remains an open problem. Intuitively, momentum is the linear combination of previous gradients, which implies that momentum has the similar direction with current adjusted gradients. This suggests that momentum is inherently compatible with our IGA. Therefore, we adjust gradients first by IGA and then update the momentum and learning rates by traditional Adam. Adaptive learning rates typically result in larger update steps for parameters (Kingma, 2014; Wilson et al., 2017; Reddi et al., 2019). For better convergence, we utilize $\lambda_{end+1}$ to balance eigenvalues in equation 5. Our extensive empirical studies in INRs (Table 8, 12, 14 in our paper) demonstrate the compatibility of IGA with the Adam optimizer.

## E. IGA on more activations

In the main text, we have evaluated our IGA on ReLU, PE (Tancik et al., 2020) and SIREN (Sitzmann et al., 2020). In this section, we apply our IGA to the most recent, meticulously crafted state-of-the-art activations with periodic variable, i.e., Gauss (Ramasinghe & Lucey, 2022), WIRE (Saragadam et al., 2023) and FINER (Zhu et al., 2024). For these activations, we follow their initial learning rates by their official guidelines. The same training strategy is adopted for these experiments. We set $end = 20$ for all these activations, which are consistent with PE and SIREN. These experiments are conducted on the first 8 images of Kodak 24. We report the average PSNR in the Table. 5. Similar to our results in the main text, our IGA consistently improves the approximation accuracy for the evaluated activations.

## F. Ablation study

As discussed in Sec. 3.4, we conduct ablation studies about the effect of sampling data and the group size $p$. All experiments are conducted on the first three images of Kodak 24 dataset, and the average PSNR values are reported for reference.

*Table 5.* Peak signal-to-noise ratio (PSNR ↑) of 2D color image approximation results on more state-of-the-art activations with periodic variable. "Gauss" and "WIRE" denote that MLPs of Gaussian and Gabor Wavelet activate function by periodic variable, respectively. The detailed settings can be found in Sec. 5.1 and Sec. E.

| Average | Gauss | | WIRE | | FINER | |
|---|---|---|---|---|---|---|
| Metric | Vanilla | **+IGA** | Vanilla | **+IGA** | Vanilla | **+IGA** |
| PSNR ↑ | 33.79 | **34.45** | 30.37 | **31.49** | 35.12 | **35.55** |

### F.1. The effect of Sampling data

In this subsection, we examine sampling data selection. We randomly sample one data from each group in each iteration, referring to the resulting MLPs as randomly sampling (RI). Additionally, we implement a variant where random sampling occurs only at the beginning, denoting these MLPs as randomly sampling initially (RSI). We also introduce sampling based on the largest residual, termed SLR. Table 6 presents the average PSNR results while maintaining the same settings as in Experiment 5.1, demonstrating that our method significantly improves performance across various sampling strategies, with SLR yielding the best average performance. Consequently, we adopt SLR for sampling strategy.

*Table 6.* Ablation experiments on the effect of sampling points. SLR denotes the sampling based on the largest residual in each iteration. RSI denotes the random sampling initially. RI denotes the random sampling in each iteration.

| Method | ReLU+IGA | PE+IGA | SIREN+IGA | Average |
|---|---|---|---|---|
| SLR | 25.44 | 34.09 | 34.85 | 31.46 |
| RSI | 25.52 | 33.77 | 33.76 | 31.02 |
| RI | 25.19 | 34.05 | 33.69 | 30.98 |
| Vanilla | 24.09 | 30.46 | 33.70 | 29.42 |

### F.2. Group size $p$ and Training time

The size $p$ of the sampling group influences the level of data similarity within the group and hardware requirements. Intuitively, a smaller $p$ reduces errors in estimating training dynamics and allows for more accurate tailored impacts on spectral bias, leading to improved representation performance. However, this leads to a larger matrix $\tilde{K}$, which results in longer run times and greater memory usage.

To further explore the effect of various $p$, we follow the MLP architecture in Experiment 5.1. All training settings are remained. We vary $p$ from $16^2$ to $128^2$. Larger sizes correspond to coarser estimates of the dynamics. We report the average training time of each iteration.

As shown in Table 7, our method under all sampling sizes has the significant improvement comparing to the baseline models. Particularly, when the size $p$ is $128^2$ that means that the size of $\tilde{K}_e$ is only $24^2$, it still improves the representation performance. That means that relatively low-precision estimates are sufficient to achieve representation improvements. Further, as $p$ decreases, representation accuracy improves; however, the extent of this improvement gradually diminishes with smaller group size $p$. Specifically, reducing the group size $p$ from $64^2$ to $32^2$ nearly doubles the average increment $\Delta_{\text{avg}}$, while the increment becomes negligible when reducing $p$ from $32^2$ to $16^2$, while the time required for $p = 16^2$ increases fivefold. This indicates that, in practical applications, there is no need to consider more precise sampling intervals; a relatively coarse partition suffices to achieve substantial improvements while maintaining efficiency.

## G. Experimental details and per-scene results of 2D color image approximation

In this section, we first include additional experimental details that were omitted in the main text due to space constraints. Then we provide the detailed per-scene results for all four metrics and more visualization results.

As illustrated in the main text, we follow the learning rate schedule of Shi et al. (2024a), which maintains a fixed rate for the first 3K iterations and then reduces it by 0.1 for another 7K iterations. For IGA, we set initial learning rates as $5e-3$ for ReLU activation and $1e-3$ for Sine activation. For all baseline models, we set initial learning rates as $1e-3$ due to poor performance observed with $5e-3$. Full-batch training is adopted.

*Table 7.* Ablation experiments on sampling group size $p$. $\Delta_{\text{avg}}$ denotes the average increment across three model architectures. "Time (ms)" denotes the average time of each iteration.

| $p$ | ReLU+IGA | PE+IGA | SIREN+IGA | $\Delta_{\text{avg}}$ | Time (ms) |
|---|---|---|---|---|---|
| $16^2$ | 25.49 | 34.25 | 34.87 | +2.12 | 256.51 |
| $32^2$ | 25.44 | 34.09 | 34.85 | +2.04 | 56.33 |
| $64^2$ | 25.19 | 32.54 | 34.10 | +1.19 | 39.76 |
| $128^2$ | 25.06 | 32.08 | 33.75 | +0.88 | 39.61 |
| Vanilla | 24.09 | 30.46 | 33.70 | – | 30.00 |

We report four metrics to comprehensively evaluate the representation performance, i.e., peak signal-to-noise ratio (PSNR), structural similarity index measure (SSIM), multi-scale structural similarity index measure (MS-SSIM) and learned perceptual image patch similarity (LPIPS), in Table 8-11. Specifically, PSNR is used to measure average representation accuracy based on MSE, while SSIM, MS-SSIM and LPIPS are more sensitive to the representation of high-frequency information. As shown in Table 8-11, all methods improve representation quality across the four metrics. However, the improvements from FR and BN are inconsistent. Although they enhance frequency-sensitive metrics to some extent, they degrade average representation accuracy and even lead to negative effects in certain cases. In contrast, our IGA method achieves consistent improvements across almost all scenarios and metrics, demonstrating the superiority of tailor impacts on spectral bias induced by our IGA. We further visualize more representation results in Fig. 7, which is consistent with our observation of Fig. 4 that mprovements of IGA are uniformly distributed across most frequency bands.

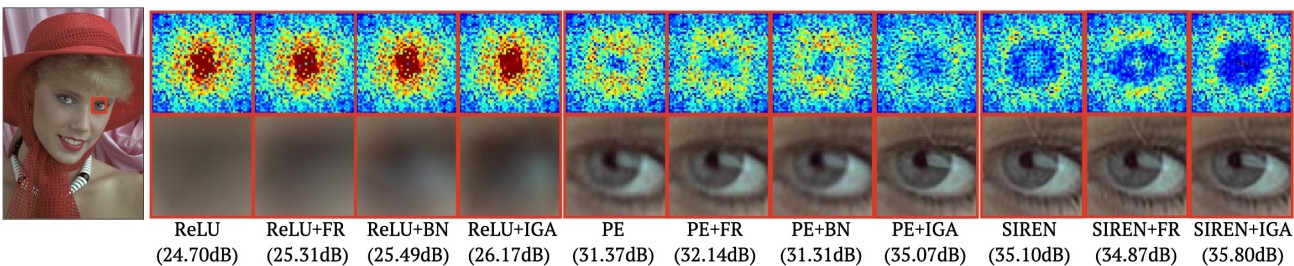

| ReLU | ReLU+FR | ReLU+BN | ReLU+IGA | PE | PE+FR | PE+BN | PE+IGA | SIREN | SIREN+FR | SIREN+IGA |
|---|---|---|---|---|---|---|---|---|---|---|
| (24.70dB) | (25.31dB) | (25.49dB) | (26.17dB) | (31.37dB) | (32.14dB) | (31.31dB) | (35.07dB) | (35.10dB) | (34.87dB) | (35.80dB) |

*Figure 7.* Visual examples of 2D color image approximation results by different training dynamics methods on Kodak 4. Enlarged views of the regions labeled by red boxes are provided. The residuals of these regions in the Fourier domain are visualized through heatmaps in the top line. The increase in error corresponds to the transition of colors in the heatmaps from blue to red. ReLU+IGA denotes that the MLPs with ReLU activations are optimized by our IGA method. Details can be found in Sec. 5.1 and Sec. G.

## H. Experimental details and per-scene results of 3D shape representation

In this section, we firstly introduce the detailed training strategy of the 3D shape representation experiment. Then we provide the intersection over union (IOU) and Chamfer distance metric values of each 3D shape and give more visualization.

Following previous works (Saragadam et al., 2023; Shi et al., 2024a; Cai et al., 2024), we train all models for 200 epochs with a learning rate decay exponentially to $0.1$ of initial rates. We set the initial learning rate as $2e-3$ for ReLU and PE; For SIREN, we set the initial learning rate as $5e-4$. FR, BN and our IGA method are adopted the same learning rate with baseline models. We typically set $end = 7$ for ReLU and PE, and $end = 14$ for SIREN. The per-scene results are reported in Table 12 and 13. It can be observed that our IGA method achieves improvements in IOU metrics across all objects, and achieves the best performance in average metrics of all objects. We visualize the Lucy object in Fig. 8.

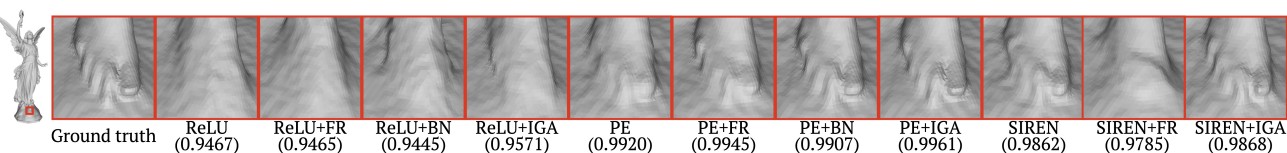

| Ground truth | ReLU | ReLU+FR | ReLU+BN | ReLU+IGA | PE | PE+FR | PE+BN | PE+IGA | SIREN | SIREN+FR | SIREN+IGA |
|---|---|---|---|---|---|---|---|---|---|---|---|
| | (0.9467) | (0.9465) | (0.9445) | (0.9571) | (0.9920) | (0.9945) | (0.9907) | (0.9961) | (0.9862) | (0.9785) | (0.9868) |

*Figure 8.* Visualization of 3D shape representation results and IOU values on the Lucy. ReLU+IGA denotes that the MLPs with ReLU activations are optimized by our IGA method. Details can be found in Sec. 5.2 and Sec. H.

# I. Experimental details and per-scene results of Learning Neural Radiance Fields

In this section, we firstly introduce some training details. Then we provide metrics of each scenario and give more visualization. As previously discussed in Experiment 5.3, we apply our method on the original NeRF model (Mildenhall et al., 2021).

Specifically, given a ray $i$ from the camera into the scene, the MLP takes the 5D coordinates (3D spatial locations and 2D view directions) of $N$ points along the ray as inputs and outputs the corresponding color $\mathbf{c}$ and volume density $\sigma$. Then $\mathbf{c}$ and $\sigma$ are combined using numerical volume rendering to obtain the final pixel color of the ray $i$. We set each ray as one group. For each group, we sample the point with the maximum integral weight. The original NeRF adopts two models: "coarse" and "fine" models. We adjust the gradients of the "fine" model as it captures more high-frequency details and remain gradients of the "coarse" model.

The "NeRF-PyTorch" codebase (Yen-Chen, 2020) is used and we follow its default settings. Besides, we also compare to previous training dynamics methods, i.e., FR and BN. Their hyperparameters follow their publicly available codes. Generally, we set the $end = 25$. For more complex scenes, such as the branches and leaves of the ficus, we set the $end = 30$ to achieve better learning of high frequencies. We report per-scene results in Table 14 and visualize several scenes in Fig. 13.

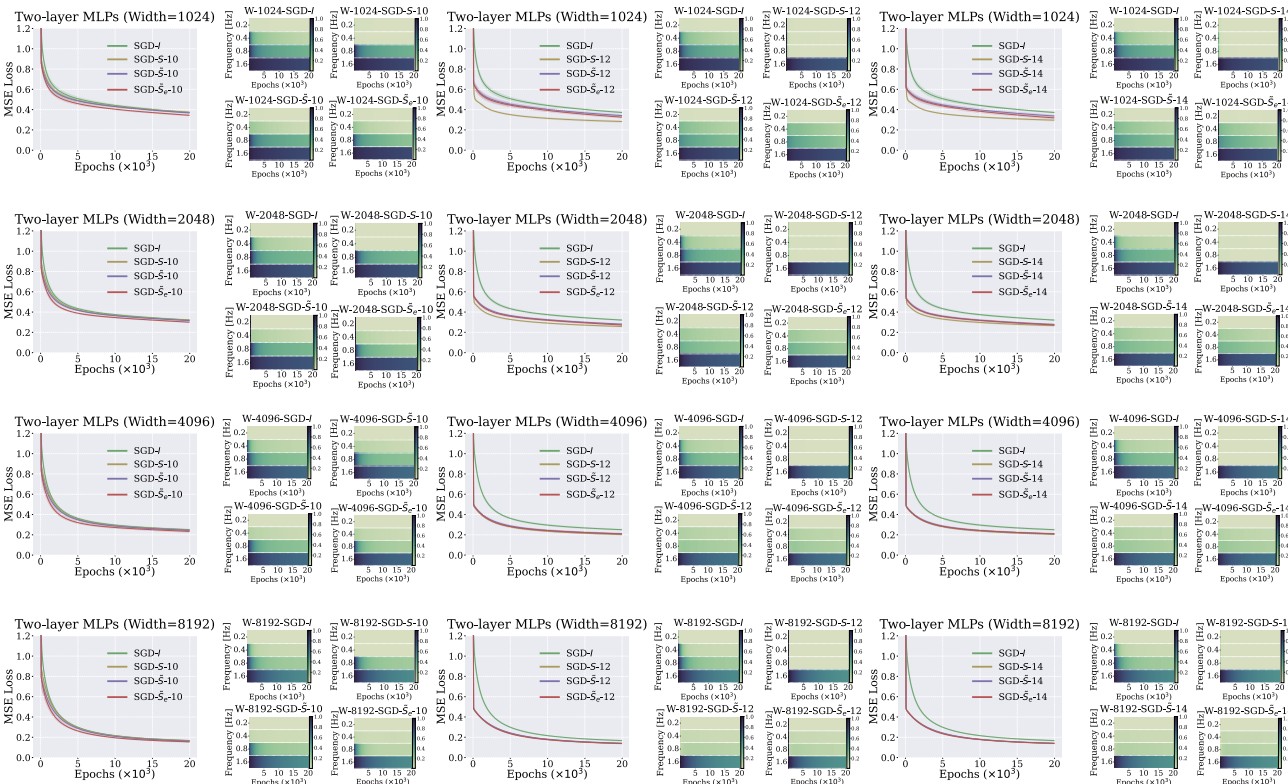

*Figure 9.* Evolution of approximation error with training iterations on time domain and Fourier domain. The shaded area indicates training fluctuations, represented by twice the standard deviation. Line plots visualize the MSE loss curves of MLPs. Heatmaps show the relative error $\Delta_k$ on four frequency bands. SGD-$\tilde{\boldsymbol{S}}_e$-10 denotes that the MLP with ReLU optimized using gradients adjusted by $\tilde{\boldsymbol{S}}_e$ with $end = 10$ and group size $p = 8$. Please zoom in for better review.

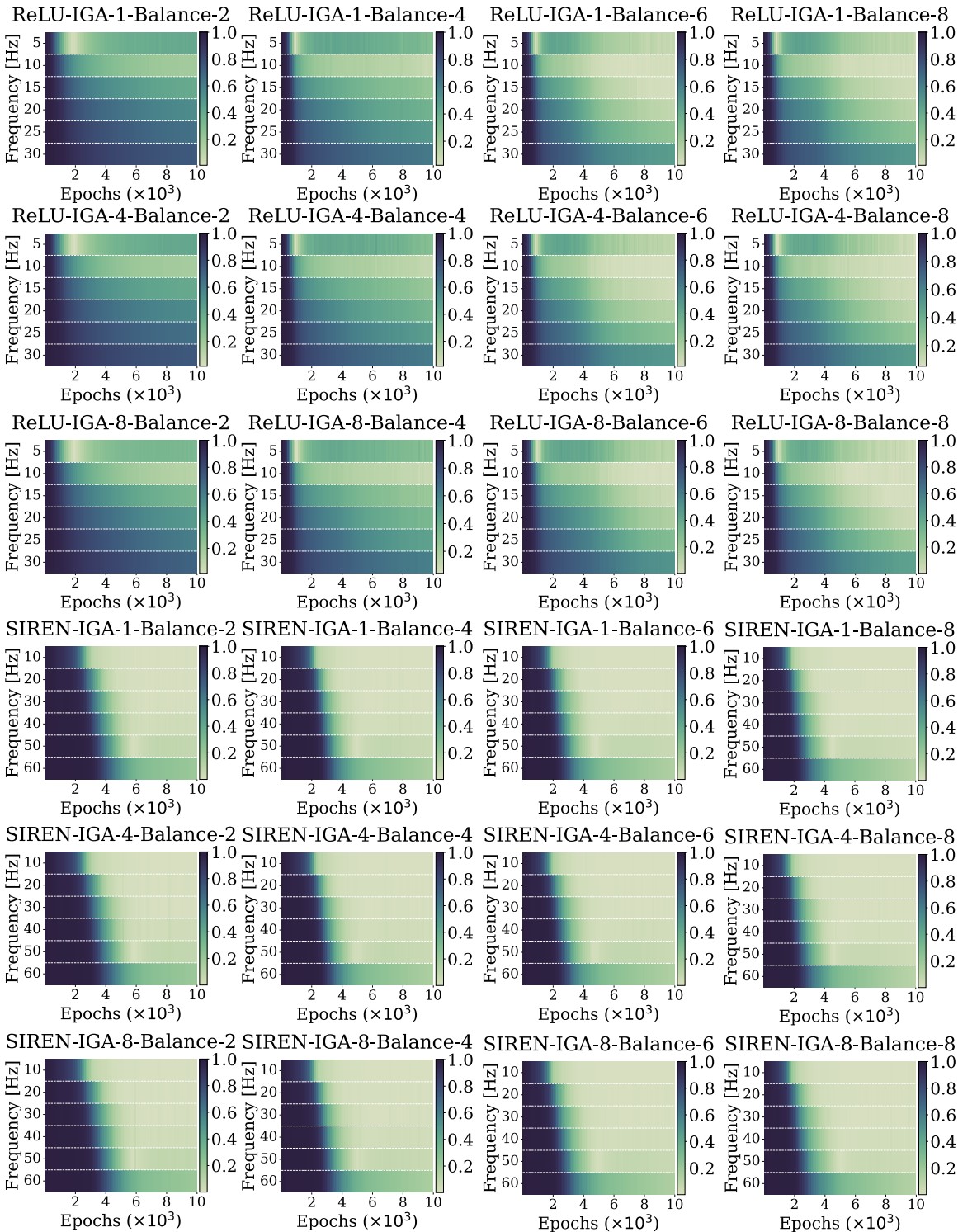

*Figure 10.* Progressively amplified impacts on spectral bias of ReLU and SIREN by increasing the number of balanced eigenvalues of $\tilde{\boldsymbol{S}}_e$ when the group size $p$ is 8. ReLU denotes that the MLP with ReLU optimized using vanilla gradients; ReLU-IGA-8-Balance-2 denotes that the MLP with ReLU optimized using gradients adjusted by $\tilde{\boldsymbol{S}}_e$ with $end = 1$ (for Adam, there are two balanced eigenvalues) and group size $p$ is 8. Details of experimental setting can be found in Sec. 4. Please zoom in for better review.

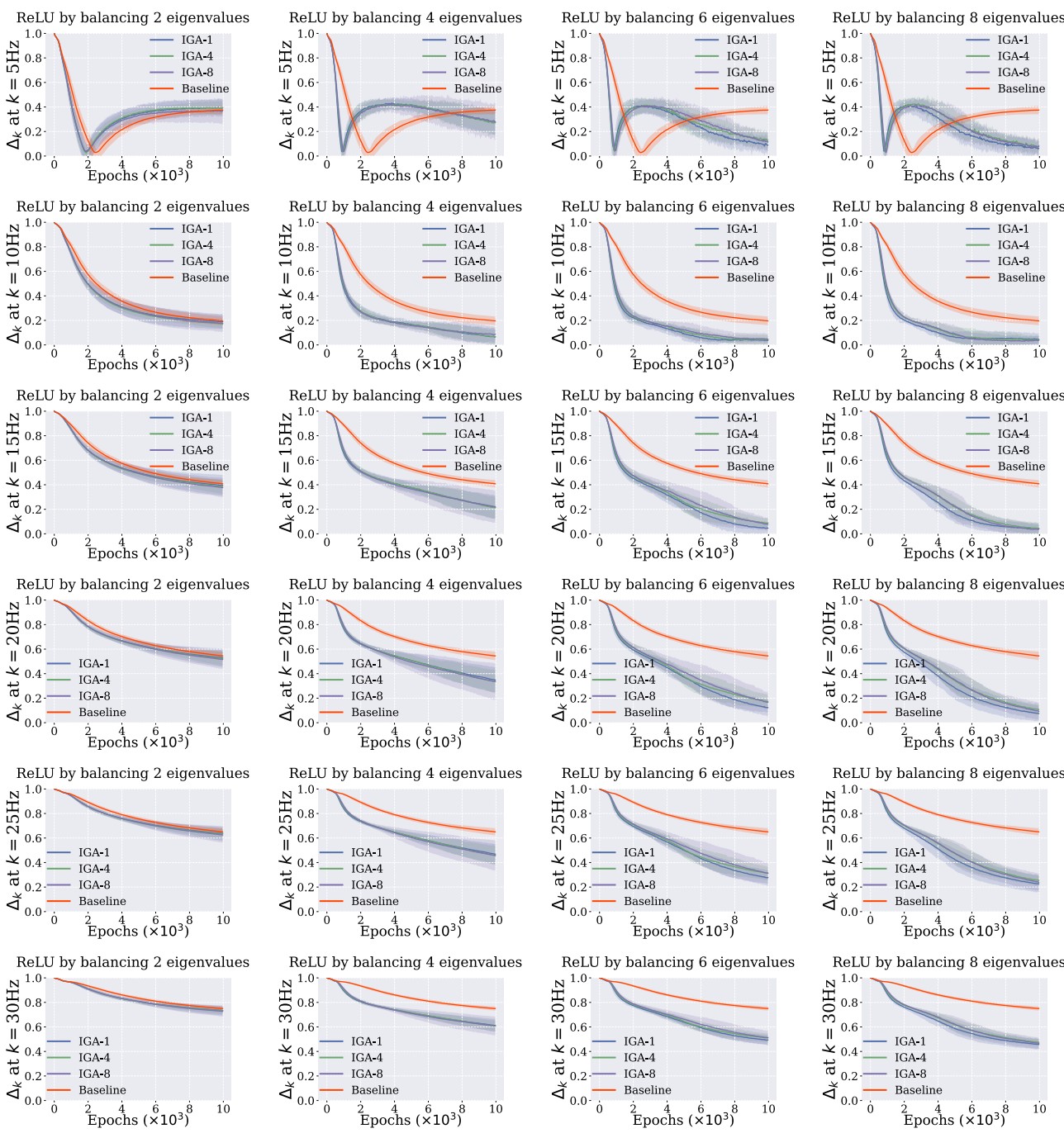

*Figure 11.* Comparison of group sizes with varying balanced eigenvalues on Relative Error $\Delta_k$ at 5, 10, 15, 20, 25, 30Hz of ReLU. IGA-1 denotes that the group size $p$ is 1, i.e., $\tilde{\boldsymbol{K}}$-based gradient adjustment; IGA-4 denotes that the group size $p$ is 4 for IGA; Baseline denotes MLPs with ReLU activations optimized by vanilla gradients. The shaded area indicates training fluctuations, represented by twice the standard deviation. Details of experimental setting can be found in Sec. 4. Please zoom in for better review.

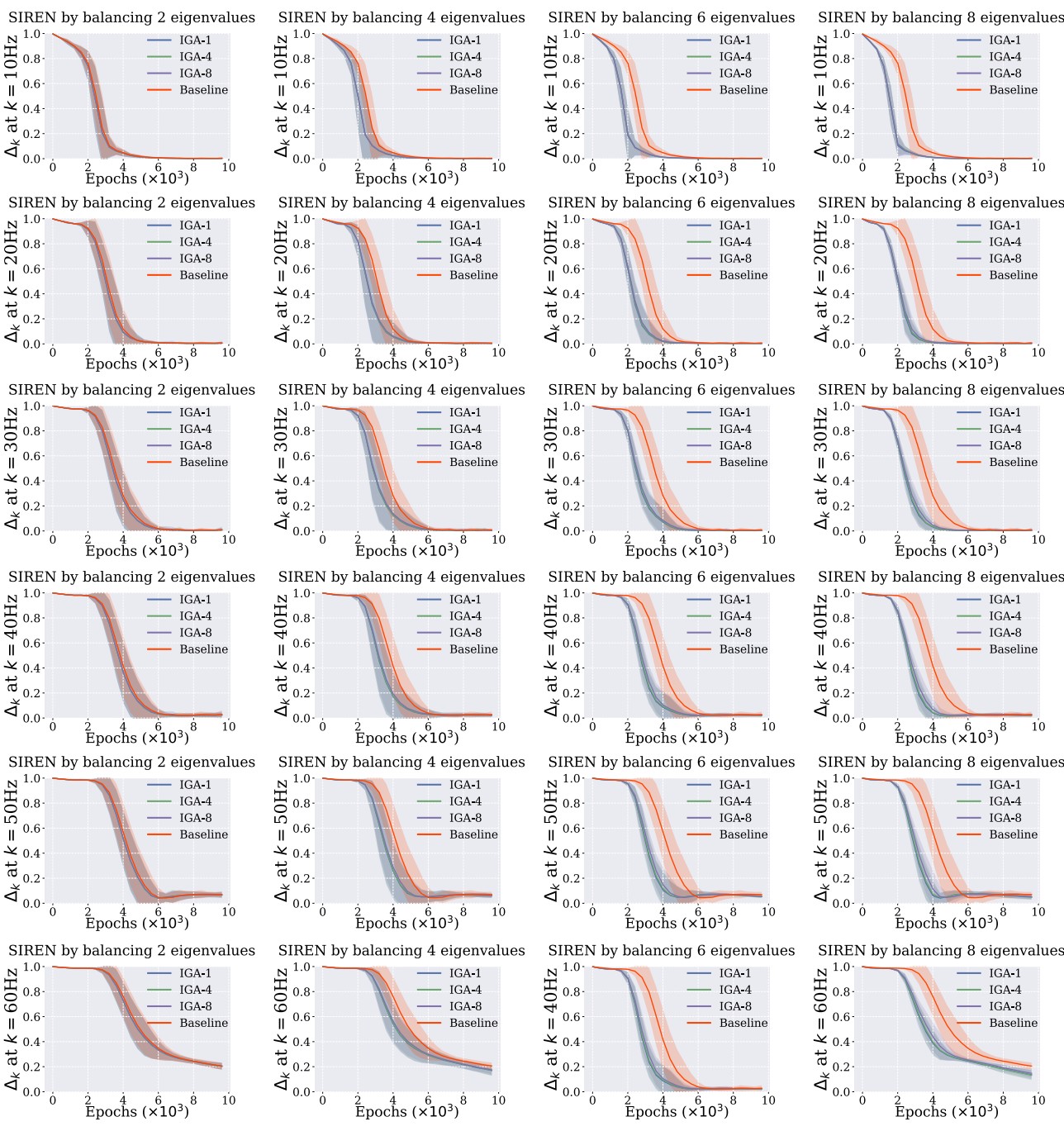

*Figure 12.* Comparison of group sizes with varying balanced eigenvalues on Relative Error $\Delta_k$ at 10, 20, 30, 40, 50, 60Hz of SIREN. IGA-1 denotes that the group size $p$ is 1, i.e., $\tilde{K}$-based gradient adjustment; IGA-4 denotes that the group size $p$ is 4 for IGA, i.e., inductive gradient adjustment; Baseline denotes the SIREN optimized by vanilla gradients. The shaded area indicates training fluctuations, represented by twice the standard deviation. Details of experimental setting can be found in Sec. 4. Please zoom in for better review.

*Table 8.* Peak signal-to-noise ratio (PSNR ↑) of 2D color image approximation results by different methods. The detailed settings can be found in Sec. 5.1.

| Method | Kodak1 | Kodak2 | Kodak3 | Kodak4 | Kodak5 | Kodak6 | Kodak7 | Kodak8 | Average |
|---|---|---|---|---|---|---|---|---|---|
| ReLU | 19.78 | 26.62 | 25.88 | 24.70 | 17.70 | 21.98 | 21.52 | 16.07 | 21.78 |
| ReLU+FR | 19.96 | 26.63 | 26.53 | 25.31 | 18.05 | 22.03 | 22.13 | 16.52 | 22.14 |
| ReLU+BN | 20.13 | 26.97 | 26.91 | 25.49 | 18.65 | 22.42 | 22.64 | 16.91 | 22.51 |
| **ReLU+IGA** | 20.42 | 27.71 | 28.20 | 26.17 | 18.81 | 22.69 | 22.96 | 16.99 | **23.00** |
| PE | 26.07 | 32.51 | 32.80 | 31.37 | 24.60 | 27.28 | 31.74 | 22.79 | 28.64 |
| PE+FR | 26.95 | 32.23 | 33.92 | 32.14 | 26.63 | 28.49 | 32.71 | 24.85 | 29.74 |
| PE+BN | 26.50 | 31.42 | 32.21 | 31.31 | 25.45 | 27.82 | 31.54 | 23.95 | 28.78 |
| **PE+IGA** | 29.17 | 35.18 | 37.91 | 35.07 | 28.06 | 31.27 | 36.60 | 26.41 | **32.46** |
| SIREN | 29.61 | 35.19 | 36.31 | 35.10 | 29.74 | 31.01 | 36.73 | 27.50 | 32.65 |
| SIREN+FR | 30.00 | 34.73 | 36.95 | 34.87 | 29.72 | 30.79 | 36.31 | 27.52 | 32.61 |
| **SIREN+IGA** | 30.10 | 35.85 | 38.60 | 35.80 | 30.13 | 31.94 | 37.68 | 27.73 | **33.48** |

*Table 9.* Structural Similarity Index Measure (SSIM ↑) of 2D color image approximation results by different methods. The detailed settings can be found in Sec. 5.1.

| Method | Kodak1 | Kodak2 | Kodak3 | Kodak4 | Kodak5 | Kodak6 | Kodak7 | Kodak8 | Average |
|---|---|---|---|---|---|---|---|---|---|
| ReLU | 0.2978 | 0.6455 | 0.7235 | 0.6199 | 0.2797 | 0.4531 | 0.5681 | 0.2785 | 0.4833 |
| ReLU+FR | 0.3063 | 0.6450 | 0.7310 | 0.6302 | 0.2920 | 0.4546 | 0.5789 | 0.2972 | 0.4919 |
| ReLU+BN | 0.3150 | 0.6461 | 0.7230 | 0.6319 | 0.3198 | 0.4635 | 0.5896 | 0.3145 | 0.5004 |
| **ReLU+IGA** | 0.3298 | 0.6628 | 0.7564 | 0.6387 | 0.3195 | 0.4753 | 0.5946 | 0.3232 | **0.5126** |
| PE | 0.7193 | 0.8126 | 0.8634 | 0.8077 | 0.7371 | 0.7553 | 0.8911 | 0.6790 | 0.7832 |
| PE+FR | 0.7653 | 0.8082 | 0.8819 | 0.8266 | 0.8074 | 0.7953 | 0.8947 | 0.7546 | 0.8167 |
| PE+BN | 0.7552 | 0.7985 | 0.8428 | 0.8198 | 0.7897 | 0.7880 | 0.8778 | 0.7525 | 0.8030 |
| **PE+IGA** | 0.8458 | 0.8806 | 0.9402 | 0.8994 | 0.8542 | 0.8778 | 0.9462 | 0.8132 | **0.8822** |
| SIREN | 0.8715 | 0.8948 | 0.9168 | 0.9060 | 0.9040 | 0.8689 | 0.9534 | 0.8643 | 0.8975 |
| SIREN+FR | 0.8781 | 0.8869 | 0.9308 | 0.9011 | 0.8971 | 0.8797 | 0.9536 | 0.8654 | 0.8991 |
| **SIREN+IGA** | 0.8830 | 0.9070 | 0.9492 | 0.9173 | 0.9110 | 0.8996 | 0.9606 | 0.8688 | **0.9121** |

*Table 10.* Multi-Scale Structural Similarity Index Measure (MS-SSIM ↑) of 2D color image approximation results by different methods. The detailed settings can be found in Sec. 5.1.

| Method | Kodak1 | Kodak2 | Kodak3 | Kodak4 | Kodak5 | Kodak6 | Kodak7 | Kodak8 | Average |
|---|---|---|---|---|---|---|---|---|---|
| ReLU | 0.5270 | 0.7839 | 0.8414 | 0.7741 | 0.4697 | 0.6534 | 0.6646 | 0.5031 | 0.6521 |
| ReLU+FR | 0.5454 | 0.7973 | 0.8621 | 0.7964 | 0.5164 | 0.6679 | 0.7024 | 0.5523 | 0.6800 |
| ReLU+BN | 0.5837 | 0.8075 | 0.8581 | 0.8047 | 0.5943 | 0.6929 | 0.7331 | 0.5976 | 0.7090 |
| **ReLU+IGA** | 0.6205 | 0.8364 | 0.8947 | 0.8249 | 0.6208 | 0.7290 | 0.7591 | 0.6211 | **0.7383** |
| PE | 0.9356 | 0.9518 | 0.9668 | 0.9489 | 0.9382 | 0.9331 | 0.9744 | 0.9242 | 0.9466 |
| PE+FR | 0.9493 | 0.9494 | 0.9713 | 0.9559 | 0.9566 | 0.9470 | 0.9755 | 0.9464 | 0.9564 |
| PE+BN | 0.9491 | 0.9507 | 0.9615 | 0.9560 | 0.9561 | 0.9481 | 0.9726 | 0.9491 | 0.9554 |
| **PE+IGA** | 0.9705 | 0.9720 | 0.9880 | 0.9788 | 0.9713 | 0.9692 | 0.9895 | 0.9622 | **0.9752** |
| SIREN | 0.9808 | 0.9817 | 0.9851 | 0.9829 | 0.9845 | 0.9714 | 0.9918 | 0.9761 | 0.9818 |
| SIREN+FR | 0.9797 | 0.9788 | 0.9877 | 0.9819 | 0.9810 | 0.9781 | 0.9926 | 0.9766 | 0.9820 |
| **SIREN+IGA** | 0.9815 | 0.9851 | 0.9895 | 0.9873 | 0.9860 | 0.9803 | 0.9921 | 0.9759 | **0.9847** |

*Table 11.* Learned Perceptual Image Patch Similarity (LPIPS ↓) of 2D color image approximation results by different methods. The detailed settings can be found in Sec. 5.1.

| Method | Kodak1 | Kodak2 | Kodak3 | Kodak4 | Kodak5 | Kodak6 | Kodak7 | Kodak8 | Average |
|---|---|---|---|---|---|---|---|---|---|
| ReLU | 0.7571 | 0.5392 | 0.4000 | 0.5547 | 0.7788 | 0.5854 | 0.6172 | 0.8089 | 0.6302 |
| ReLU+FR | 0.7692 | 0.5513 | 0.3871 | 0.5362 | 0.7829 | 0.6110 | 0.6131 | 0.8008 | 0.6315 |
| ReLU+BN | 0.7654 | 0.5471 | 0.3865 | 0.5458 | 0.7188 | 0.5959 | 0.5859 | 0.7999 | 0.6182 |
| **ReLU+IGA** | 0.6857 | 0.4770 | 0.2927 | 0.4730 | 0.6695 | 0.5609 | 0.5278 | 0.7527 | **0.5549** |
| PE | 0.2803 | 0.2092 | 0.1062 | 0.2128 | 0.2643 | 0.2517 | 0.1218 | 0.3322 | 0.2223 |
| PE+FR | 0.2547 | 0.2278 | 0.0905 | 0.1897 | 0.1566 | 0.2179 | 0.1154 | 0.2426 | 0.1869 |
| PE+BN | 0.3161 | 0.2605 | 0.1474 | 0.2065 | 0.2393 | 0.2592 | 0.1554 | 0.2926 | 0.2346 |
| **PE+IGA** | 0.1363 | 0.0938 | 0.0256 | 0.0956 | 0.1056 | 0.1068 | 0.0310 | 0.1559 | **0.0938** |
| SIREN | 0.1052 | 0.0878 | 0.0661 | 0.0994 | 0.0654 | 0.0967 | 0.0271 | 0.0980 | 0.0807 |
| SIREN+FR | 0.0928 | 0.0888 | 0.0465 | 0.1121 | 0.0661 | 0.1258 | 0.0317 | 0.0865 | 0.0813 |
| **SIREN+IGA** | 0.0895 | 0.0746 | 0.0246 | 0.0858 | 0.0628 | 0.0853 | 0.0199 | 0.0921 | **0.0668** |

*Table 12.* Intersection over Union (IOU) of 3D shape representation by different methods. The detailed settings can be found in Sec. 5.2 and Appendix H.

| Method | Thai | Armadillo | Dragon | Bun | Lucy | Average |
|---|---|---|---|---|---|---|
| ReLU | 0.9379 | 0.9756 | 0.9708 | 0.9924 | 0.9467 | 0.9647 |
| ReLU+FR | 0.9428 | 0.9756 | 0.9707 | 0.9914 | 0.9465 | 0.9654 |
| ReLU+BN | 0.9416 | 0.9789 | 0.9727 | 0.9934 | 0.9445 | 0.9662 |
| **ReLU+IGA** | 0.9563 | 0.9805 | 0.9793 | 0.9931 | 0.9571 | **0.9733** |
| PE | 0.9897 | 0.9956 | 0.9953 | 0.9985 | 0.9920 | 0.9942 |
| PE+FR | 0.9929 | 0.9979 | 0.9964 | 0.9990 | 0.9945 | 0.9961 |
| PE+BN | 0.9916 | 0.9968 | 0.9954 | 0.9985 | 0.9907 | 0.9946 |
| **PE+IGA** | 0.9943 | 0.9979 | 0.9975 | 0.9990 | 0.9961 | **0.9970** |
| SIREN | 0.9786 | 0.9908 | 0.9937 | 0.9953 | 0.9862 | 0.9889 |
| SIREN+FR | 0.9731 | 0.9935 | 0.9905 | 0.9972 | 0.9785 | 0.9866 |
| **SIREN+IGA** | 0.9802 | 0.9920 | 0.9938 | 0.9958 | 0.9868 | **0.9897** |

*Table 13.* Chamfer Distance of 3D shape representation by different methods. The detailed settings can be found in Sec. 5.2 and Appendix H.

| Method | Thai | Armadillo | Dragon | Bun | Lucy | Average |
|---|---|---|---|---|---|---|
| ReLU | 5.6726e-06 | 5.5805e-06 | 6.5588e-06 | 8.0611e-06 | 3.8048e-06 | 5.9356e-06 |
| ReLU+FR | 5.2954e-06 | 5.6664e-06 | 6.5062e-06 | 8.2686e-06 | 3.5963e-06 | 5.8666e-06 |
| ReLU+BN | 5.3489e-06 | 5.4240e-06 | 5.8546e-06 | 7.8738e-06 | 3.5496e-06 | 5.6102e-06 |
| **ReLU+IGA** | 4.7463e-06 | 5.4097e-06 | 5.7295e-06 | 8.2330e-06 | 3.3144e-06 | **5.4866e-06** |
| PE | 4.2759e-06 | 5.2162e-06 | 5.3788e-06 | 7.7409e-06 | 3.0036e-06 | 5.1231e-06 |
| PE+FR | 4.2272e-06 | 5.2246e-06 | 5.3514e-06 | 7.7974e-06 | 2.9542e-06 | 5.1109e-06 |
| PE+BN | 4.2463e-06 | 5.2030e-06 | 5.4122e-06 | 7.7698e-06 | 2.9493e-06 | 5.1161e-06 |
| **PE+IGA** | 4.2551e-06 | 5.1524e-06 | 5.4319e-06 | 7.7695e-06 | 2.9288e-06 | **5.1076e-06** |
| SIREN | 4.2920e-06 | 7.8573e-06 | 5.3923e-06 | 7.9121e-06 | 2.9870e-06 | 5.6881e-06 |
| SIREN+FR | 4.3588e-06 | 5.1830e-06 | 5.4719e-06 | 7.7805e-06 | 3.0217e-06 | 5.1632e-06 |
| **SIREN+IGA** | 4.3520e-06 | 5.2203e-06 | 5.3757e-06 | 7.8457e-06 | 2.9934e-06 | **5.1574e-06** |

*Table 14.* Per-scene results of learning 5D neural radiance fields by different methods. Details can be found in Sec. 5.3 and Appendix I.

| | Methods | Ship | Materials | Chair | Ficus | Hotdog | Drums | Mic | Lego | Average |
|---|---|---|---|---|---|---|---|---|---|---|
| PSNR ↑ | NeRF | 29.30 | 29.55 | 34.52 | 29.14 | 36.78 | 25.66 | 33.37 | 31.53 | 31.23 |
| | NeRF+FR | 29.50 | 29.70 | 34.54 | 29.35 | 37.08 | 25.74 | 33.35 | 31.55 | 31.35 |
| | **NeRF+IGA** | 29.49 | 29.87 | 34.69 | 29.38 | 37.22 | 25.80 | 33.48 | 31.83 | **31.47** |
| SSIM ↑ | NeRF | 0.8693 | 0.9577 | 0.9795 | 0.9647 | 0.9793 | 0.9293 | 0.9783 | 0.9626 | 0.9526 |
| | NeRF+FR | 0.8729 | 0.9600 | 0.9797 | 0.9665 | 0.9792 | 0.9304 | 0.9786 | 0.9626 | 0.9537 |
| | **NeRF+IGA** | 0.8716 | 0.9619 | 0.9807 | 0.9667 | 0.9801 | 0.9315 | 0.9791 | 0.9648 | **0.9546** |
| LPIPS → | NeRF | 0.077 | 0.021 | 0.011 | 0.021 | 0.012 | 0.052 | 0.022 | 0.019 | 0.029 |
| | NeRF+FR | 0.071 | 0.020 | 0.011 | 0.019 | 0.013 | 0.050 | 0.021 | 0.019 | 0.028 |
| | **NeRF+IGA** | 0.074 | 0.019 | 0.010 | 0.019 | 0.011 | 0.049 | 0.020 | 0.017 | **0.027** |

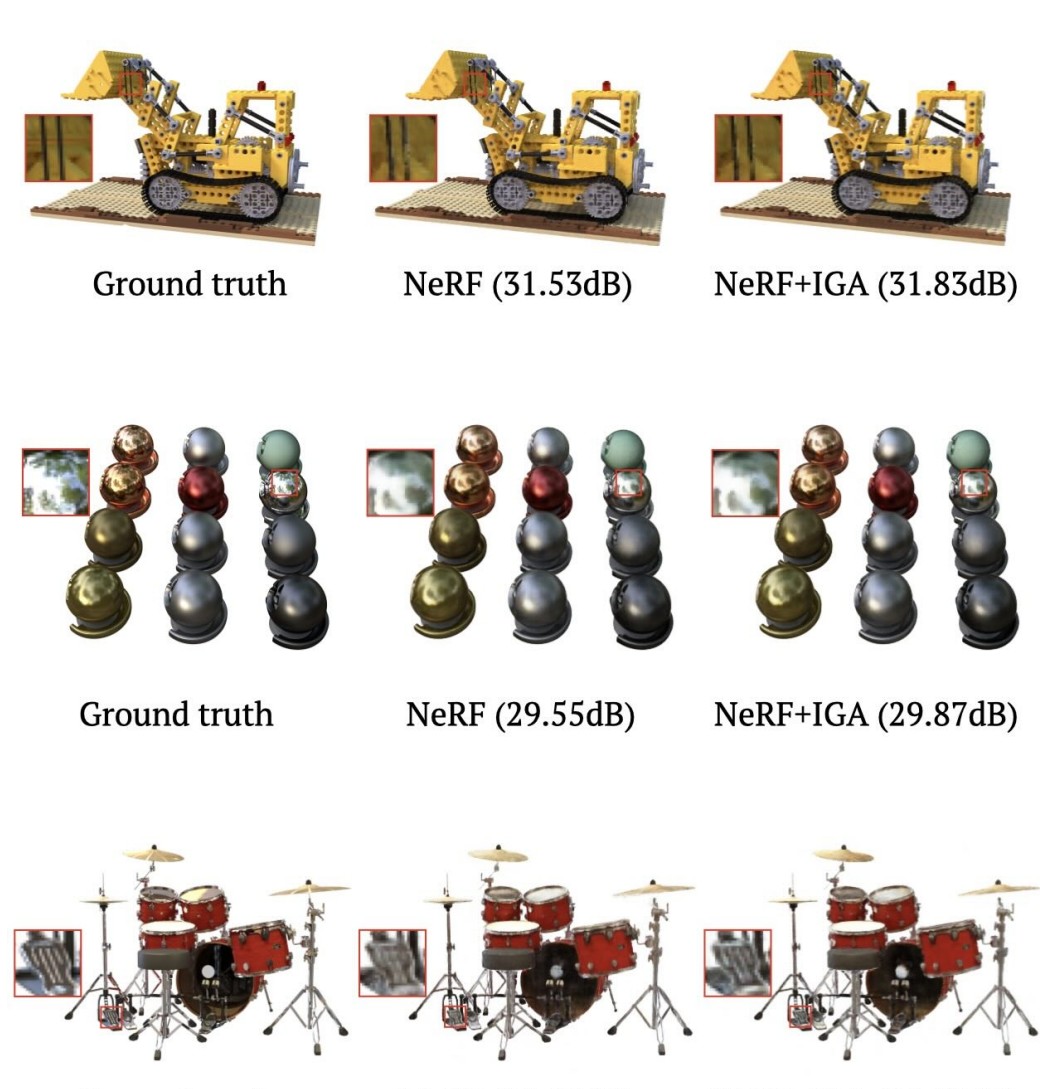

*Figure 13.* Visual examples of novel view synthesis results of NeRF and NeRF+IGA. Details can be found in Sec. 5.3 and Appendix I.

