# OpenReview forum: "Inductive Gradient Adjustment for Spectral Bias in Implicit Neural Representations"
_ICML.cc/2025/Conference — ICML 2025 poster_

### Official Review · Reviewer_ARPB · 2025-03-10

**Overall Recommendation:** 3

**Summary:**

This paper introduces a practical Inductive Gradient Adjustment (IGA) method to address spectral bias in Implicit Neural Representations (INR) by using inductive generalization of the eNTK-based gradient transformation matrix. The effectiveness of IGA is evaluated across a wide range of INR tasks, and a theoretical justification for its impact on spectral bias is also provided.

**Claims And Evidence:**

The claims are well-articulated and supported by compelling evidence.

**Essential References Not Discussed:**

The references are comprehensive.

**Experimental Designs Or Analyses:**

I have reviewed the experimental design and analyses, and most aspects are well-presented. However, there is limited discussion regarding training time. The use of the eNTK-based gradient transformation matrix to address spectral bias in INR involves computing the eNTK, which is computationally expensive, as noted in Section 3.2. It is important to address this issue and consider potential solutions, such as INT [1] (“Nonparametric teaching of implicit neural representations”).

**Methods And Evaluation Criteria:**

The proposed methods and evaluation criteria are logical and well-founded.

**Other Comments Or Suggestions:**

- There is an extra parenthesis in Theorem 3.1; “(max(gi(λ))” should be written as “max(gi(λ))”.

- The explanation of spectral bias could be clearer, similar to the one provided in [2] ("Fourier features enable networks to learn high-frequency functions in low-dimensional domains").

- The term gi(λi) should be defined prior to its introduction in Equation (2).

**Other Strengths And Weaknesses:**

**Strengths**:

- The paper is well-structured and easy to follow.

- The evaluation of IGA is thorough across a wide range of INR tasks.

- The theoretical explanation of IGA's impact on spectral bias is also provided.

**Weaknesses**:

- The notation for $\Theta$ in Theorem 3.2 is unclear.

- There is limited discussion regarding training time.

**Questions For Authors:**

No.

**Relation To Broader Scientific Literature:**

This paper aims to enhance the accuracy of Implicit Neural Representations (INR), which could be advantageous for fields related to INR.

**Theoretical Claims:**

I have reviewed the proof in this paper, and it appears to be correct.

---

> ### Author Rebuttal · Authors · 2025-04-01
>
> # Discussion on training time
> ***
> Thanks for your overall recognition of our method, experimental design, and analysis. Following your suggestion, we provide a further discussion about training time of our method as a supplement to Table 6 in our Appendix. Please refer to our response **“Time and memory analysis”** to Reviewer 7Wmr. The complete results can be found in: https://anonymous.4open.science/r/iga-inr-icml2025-3A4D/time_memory.md. Overall, IGA averages 1.56× the training time and 1.34× the memory of the baseline but achieves greater improvements (on average, **2.0× those of prior FR and BN**, as shown in the main text). Compared to the vanilla adjustment method by full eNTK matrix, **IGA saves at least 1/4 of the training time**. Thus, IGA improves fitting accuracy by enhancing training dynamics without significantly increasing time.
>
> The additional training time overhead comes from the matrix decomposition. As you mentioned, computing the eNTK and performing eigenvalue decomposition for the population data is intractable due to both memory consumption and computational complexity. Thanks to our theoretical foundations (Theorem 3.1 & 3.2), we only need to decompose the eNTK matrix of the sampled data and inductively generalize them to the population data, which allows for a significant reduction in computational cost while maintaining the effectiveness of spectral bias mitigation. Table 6 in our Appendix shows that a small amount of the sampled data can also lead to a nontrivial improvement.
>
> # Discussion on potential solutions
> ***
> Thanks for your constructive advice. We acknowledge INT as a potentially insightful approach and have cited it in the line 014 (right column) of our paper. We will include a following discussion of such methods in the final version. However, the problems that INT aims to address and its use of eNTK are different from ours. INT aims to address **the costly training of INRs via the sample selection**, while our IGA aims to **improve the spectral bias of INRs via the training dynamics adjustment**. Based on the nonparametric teaching perspective, the eNTK in INT is used to compute the functional gradient；samples with larger functional gradients are prioritized for selection. **Therefore, the specific values are not required for INT, only the relative magnitudes matter.** They found that the functional gradient is positively correlated with the discrepancy between the MLP output and the target function; thus, direct computation of the eNTK can be avoided.
>
> While, for IGA, based on the linear dynamics perspective, eigenvalues of the eNTK matrix control the convergence speeds of the MLP on the corresponding eigenvectors; the uneven distribution of eigenvalues leads to varying convergence speeds across feature directions, which is considered as one of the potential causes of spectral bias. IGA aims to uniform these eigenvalues of the underlying eNTK matrix during training to balance the convergence speeds and overcome spectral bias. **Therefore, explicit computation of specific eigenvalues and the corresponding eigenvectors on the eNTK matrix is required for IGA.**
>
> As you mentioned that the computational cost of the eNTK is enormous, IGA adopts the inductive generalization of the eNTK-based gradient transformation matrix estimated from the sampled data. This avoids the computation of the eNTK matrix for the entire dataset, significantly reducing computational overhead, while still providing a non-trivial improvement. Compared to the vanilla method, i.e, full eNTK matrix, this **saves at least 1/4 of the training time**, as mentioned in the answer "Discussion on training time".
>
> # Response for writing-related comments
> ***
> Thanks for your patient review of our theorem and proofs. $\Theta(\cdot)$ in Theorem 3.2 of the main text denotes a tight bound. Concretely, we take the $\epsilon_2=\Theta(m^{-3/2})$ in line 215, right column as an example. $\epsilon_2=\Theta(m^{-3/2})$ indicates that as the variable $m$ grows large, $\epsilon_2$ behaves asymptotically like $m^{-3/2}$ up to constant factors. That is, as stated in the formal version of Theorem 3.2 (in Section A.4 of the Appendix), $\epsilon_2=\frac{\eta k^3 R_0^2}{m^{3/2}}$, where  $\eta, k, R_0$ are constants. Following your suggestions, we will remove the extra parenthesis in Theorem 3.1 (line 186) for readability, define $g_i(\lambda)$ before Equation 2, and refine our explanation of spectral bias for clarity.

---

### Official Review · Reviewer_x2oC · 2025-03-13

**Overall Recommendation:** 3

**Summary:**

The goal of the paper is to mitigate spectral bias in implicit neural representations by changing the training dynamics. The paper considers the well-known connection between the neural tangent kernel (NTK) and the linear training dynamics, which reproduces spectral bias through the eigenvalues of the NTK matrix. The main idea is to introduce a preconditioning matrix into the gradient updates during training, where the preconditioner is designed to compensate for the spectral bias in the NTK. There are two theoretical contributions required to realize this idea practically: (1) showing that the empirical NTK is a good approximation of the NTK as network width increases, since the empirical NTK is more practical to compute, and (2) showing a method to approximate the empirical NTK via batchwise computations that are computationally tractable even for large datasets. Combining these theoretical contributions, the paper proposes a practical algorithm called Inductive Gradient Adjustment (IGA), that can be applied to existing INR architectures to mitigate spectral bias during training.

**Claims And Evidence:**

Of the three bullet point contribution claims at the end of the introduction, the first two seem redundant with each other (ie. could be combined into one bullet). An alternative suggestion would be to have these two claims be separated into a theoretical contribution and an algorithmic contribution, since (as described in the summary section of the review) there are interesting theoretical contributions that can be separated from (but help justify and make practical) the algorithmic idea of using a preconditioner matrix to correct the spectrum of the NTK during training.

The theoretical claims are well validated by the first two toy experiments, and the improvements on the image fitting task are compelling real-world validation.

**Essential References Not Discussed:**

N/A

**Experimental Designs Or Analyses:**

The first experiment is a toy 1D signal fitting experiment (in Figure 1) to verify the theoretical claims. In this toy setting where the analytical NTK and empirical NTK are both tractable, this experiment shows that the proposed NTK approximation does provide similar performance to the empirical NTK, which itself does slightly worse but still decently similarly compared to the analytical NTK. This experiment also shows that the approximation quality improves as network width increases from 1024 to 4096.

The second experiment (Figures 2 and 3) also uses a 1D toy function with known frequency decomposition, and validates the theoretical prediction that changing the number of NTK eigenvalues that are adjusted does have qualitatively the expected impact on spectral bias.

The remaining experiments show compelling improvements on 2D image fitting and more incremental improvements on 3D shape and radiance field fitting tasks.

**Methods And Evaluation Criteria:**

Overall I find the methods and evaluation criteria are clear and well thought out. The toy experiments do validate the theory, and the image fitting results are compelling. However, the 3D shape and radiance field results are very similar to prior work, with only a marginal improvement.

**Other Comments Or Suggestions:**

The theoretical contribution is, in my view, stronger than the introduction of the paper led me to expect. In particular, the idea of approximating the empirical NTK to make it practical as a gradient preconditioner is clever and elegant, so I’d like to see it mentioned earlier.

Overall the writing is straightforward to follow, but some of the writing would benefit from copy editing. For example, the last sentence of the abstract might intend to say “tailoring the impacts of spectral bias” instead of “tailored impacts on spectral bias”? There is also some opportunity to tighten up the writing and remove redundancy. For example, in the left column of page 2 there are 6 sentences that describe “purposefully” overcoming spectral bias by adjusting the empirical NTK spectrum. On line 120 (left column), “improve” should be “improves.”

Section 3.1 is a helpful and well-written introduction to spectral bias from an NTK perspective. I might suggest separating it into a “background” section rather than having it as part of the “method” section, since it is a summary of important background rather than a novel contribution.

Figure 1 caption: “on time” should be “in time.”

Figure 3: The line colors are difficult to distinguish without zooming in (particularly the blue and purple lines). The lines might be more easily distinguished either by changing colors or by adding some variation in line style (e.g. dotted, dashed, markers).

Figure 4: The result is compelling, but to improve presentation I suggest (1) including the ground truth image, and (2) including the acronym definitions in the caption.

**Other Strengths And Weaknesses:**

N/A; discussed in the other sections of the review.

**Questions For Authors:**

It would be helpful to provide a bit of exposition about the g_i below equation 2. Are these free parameters to be chosen when constructing the preconditioning matrix S, or are they constrained or derived in some way? Also, please refer to some questions embedded in other parts of the review.

**Relation To Broader Scientific Literature:**

The second paragraph of the introduction draws a distinction between methods that mitigate spectral bias through model architecture versus through training procedure, and seems to imply that it’s simpler or easier to modify training procedure. I’m not sure why this would be true; many of the existing architecture-based INRs use very simple changes to a standard MLP, often just changing the activation function (e.g. SIREN, WIRE) or adding an input embedding (e.g. Fourier features). Later in the introduction the authors make the point that their proposed IGA training strategy can be used in conjunction with these architectural modifications for further benefit; to me this is a stronger justification for the proposed method.

**Theoretical Claims:**

I appreciate the structure of how the theoretical contributions are introduced. Section 3.2 gives a clear overview of what the theoretical contributions are and how they will be put together to create the IGA training procedure.

That said, the informal statement of Theorem 3.1 is a bit confusing in that it’s not entirely clear where the assumptions end and the statement of the result begins. It’s also not entirely clear where the functions g_i come from; can they be chosen at will from the set of all Lipschitz functions?

The informal statement of Theorem 3.2 also uses some notation that is not entirely clear. In particular, there is a statement “...such that [quantity A in absolute value], [quantity B in norm] < epsilon.” Does this mean that max(quantity A in absolute value, quantity B in norm) < epsilon? I’m also a bit confused why there are brackets (starting at p and ending at r_e) in the main result, since if these were removed (and if p were reordered) then we would have exactly the j_i’th row of \tilde K_e in the expression, which might be clearer to read.

---

> ### Author Rebuttal · Authors · 2025-04-01
>
> # Answer for 3D shape and radiance field results
> ***
> Thanks for recognizing our experiments and evaluation approach. We further analyze IGA by comparing it with Fourier Reparameterization (FR) and Batch Normalization (BN), two key training dynamics adjustment methods for mitigating INR spectral bias. For 3D shape task, compared to the vanilla baseline models, FR and BN achieve an average IoU improvement of $9 \times 10^{-4}$ and $6 \times 10^{-4}$, respectively, across five scenes. IGA further achieves an IoU improvement of $4.1 \times 10^{-3}$, which is on average **5$\times$ that of FR and BN**. For the neural radiance field, FR and BN achieve 0.12dB and 0.14dB in PSNR, respectively. The average improvement of IGA is up to 0.24dB, which is nearly **2$\times$ that of FR and BN**, as mentioned in line 417-418, right column. Please note that the improvements achieved by all these methods are non-trivial, especially considering that they are achieved without altering the network inference structure or inference speed.
>
> # Clarification of theorems
> ***
> Thanks for recognizing our theoretical contributions and careful reading. We apologize for any ambiguity in the statement. For Theorem 3.1, there are two assumptions. The first one is that $v_i^{\top}\tilde{v}_i >0$. This assumption is easy to satisfy in practice, as both $\tilde{K}$ and $K$ are real symmetric matrices, and $\tilde{K}$ gradually converges to $K$, leading to increasingly similar eigenvectors. The second one is that $\eta < ( ) $. This assumption is made to satisfy the requirements of the linear dynamic analysis. For Theorem 3.2, the statement is indeed meant to convey that max(quantity A in absolute value, quantity B in norm) < epsilon. As you suggested, removing the brackets '[]' and reordering $p$ would make the expression clearer and exactly result in the $j_i$-th row of $\tilde{K}_e$. **We will incorporate the above suggestions into the final version.**
>
> # Discussion about the Lipschitz functions $g_i$
> ***
> Thanks again for careful reading of our theorems and insightful comments. We will revise the Eq. 2 to provide more exposition of $g_i$. Initially, the functions $g_i, i=1,..., N$ are introduced to represent the potential eigenvalue adjustment functions. During the analysis, we found that **a Lipschitz continuity condition on these functions serves as a simple yet effective condition for the equivalence between $\tilde{K}$-based and $K$-based adjustments**. This condition ensures that, for a set of N Lipschitz functions, there exists a specific Lipschitz constant upper bound, allowing for the existence of a width $m$ such that all adjusted eNTK eigenvalues converge to their corresponding potential adjusted NTK eigenvalues.  Following this theoretical analysis, In practice, we simply map the largest $n$ eigenvalues of the $\tilde{K}$ to the $(n+1)$st largest eigenvalue of $\tilde{K}$, keeping the remaining eigenvalues unchanged (please refer to Eq. 5 in our paper). The corresponding operation for the potential corresponding $K$ is mapping the largest $n$ eigenvalues of the $K$ to the $(n+1)$st largest eigenvalue of $K$, and others remaining unchanged. In this setting, the potential errors of these mapped $n$ eigenvalues is dominated by the $(n+1)$st largest eigenvalue. As the eigenvalues from the $(n+1)$-th largest to the smallest are kept unchanged, it follows that their $g_i$functions are the Identity function, with a Lipschitz constant equal to 1. Our wide results and empirical analysis show that the aforementioned simple $g_i$  setting effectively balances the convergence speeds of different frequency components.
>
> # Answer for relation to Broader Scientific Literature
> ***
> Thanks for your thoughtful comments. The second paragraph indeed aims to highlight the distinction between training dynamics-based methods and architecture-based approaches such as SIREN and WIRE. As you pointed out, our method can be combined with architectural modifications for further benefits, providing a stronger justification. We will revise this in the final version.
>
>
> # Refinements on Writing
> ***
> Thanks for your valuable writing suggestions. As you pointed out, the first two contributions at the end of the introduction may appear somewhat redundant. We appreciate your suggestion to distinguish **theoretical and algorithmic contributions**. We will revise the manuscript to better highlight our theoretical insights. Follow your suggestions, we will **highlight the empirical NTK approximation earlier** and restructure Section 3.1 into a **“Background” section**. Fig. 9 will be **resized** to match Fig. 10, and **ground truth** will be added to Fig. 4. **Abbreviation definitions** will be included in captions, and **line distinctions** in Fig. 3 will be adjusted. Minor typos and redundancies such as "improves" will also be carefully revised.

---

### Official Review · Reviewer_7Wmr · 2025-03-14

**Overall Recommendation:** 4

**Summary:**

The paper presents a Neural Tangent Kernel-based approach to improving the spectral bias of implicit neural representations, which have been shown to be biased towards low frequencies. The paper summarizes the NTK theory and proposes a way to estimate the K matrix using a subset of the training samples, making this estimation tractable, compared to computing the actual matrix. This matrix can then be used, similarly to previous work, to steer the training of INRs into having less bias. Theoretical guarantees for the estimation are provided, as well as experiments. The experiments are: two simple experiment on function approximation, useful for an analysis on the proposed approach but not for drawing practical conclusions; an experiment on image representation; an experiment on 3D shape representation; an experiment on neural radiance fields. All the experiments show that, in the proposed settings, the method can be applied to existing methods and outperform them.

##Update after rebuttal
The rebuttal addressed most of my concerns, especially the ones about practicality in terms of time and memory requirements. I still believe an auto decoder-based 3D shape experiment would be more effective and useful than the one proposed, but overall I propose acceptance of the paper.

**Claims And Evidence:**

The claims are mostly supported by the shown evidence. The only exception, in my opinion, being the claim at page 2 line 97, where the proposed method is labeled as "practical". While the analyses and experiments indeed show that the method works in the tested scenarios and with the tested baselines, for the method to be convincingly practical an analysis on its additional time and memory requirements would be needed. A partial analysis is provided in the supplementary, table 6. However it is only shown for one experiments, it does not mention memory requirements, and the time impact is shown to be highly sensitive to the choice of p. I believe a more detailed analysis should be provided for all experiments, with data about time and memory requirements being added to the tables in the main paper.

**Essential References Not Discussed:**

To the best of my knowledge, no, except for the additional experimental methods mentioned above.

**Experimental Designs Or Analyses:**

Yes. The experiments are mostly well designed, and they include several standard scenarios, baselines and metrics, which overall are convincing about the potential of the proposed approach.
E1) As mentioned above, a time and memory requirements analysis is required to support the claim about practicality, but it's currently very limited and relegated to the appendix.
E2) The methods tested in the main paper (ReLU MLP, MLP with PE, Siren, Vanilla Nerf) do show that the proposed approach improves on vanilla approaches, however more recent and complex methods have been proposed, and a few of them should be used to convince the reader about the practical usability of the proposed framework. The supplementary additionally shows Gauss, WIRE and FINER for image representation. MFN[1]/Bacon[2] should be considered too, as well as for shape representation, and a more recent and powerful NeRF model should be added as well. I believe these results belong to the main paper, as they serve to support the claims made by the authors.

E3) Additionally, the experiment on 3D shape representation is not fully convincing for 2 reasons. 1) While IOU is an important metric, Chamfer Distance should be reported as well. In my experience they can behave differently, and they are both considered standard in this field. 2) As far as I understand, a simple setting is shown where each network is overfitted to a single shape. This is not considered a very useful scenario in practice, since INRs are usually used in an auto-decoder fashion to learn multiple shapes. In the auto-decoder scenario, different methods can actually behave quite differently, so this would be a much more useful experimental setting.

[1] Multiplicative Filter Networks, Rizal Fathony, Anit Kumar Sahu, Devin Willmott, J Zico Kolter, ICLR 2021
[2] BACON: Band-limited Coordinate Networks for Multiscale Scene Representation, David B. Lindell, Dave Van Veen, Jeong Joon Park, Gordon Wetzstein

**Methods And Evaluation Criteria:**

Yes, the method seems well grounded and justified, providing tangible benefits.

**Other Comments Or Suggestions:**

These did not impact the review:
Page 1 Line 33-35 column 2: it's not clear which methods have an impact on complexity. Some of the mentioned ones don't, as far as I know (such as SIREN)
Page 6 Line 322 column 1: "fixed last layer in" -> "fixed last layer as in"?
Page 6 Line 298-301 column 2: I've found the phrasing confusing

**Other Strengths And Weaknesses:**

Strengths:
S1) The proposed method seems original and well grounded in its analysis
S2) The paper is well written and easy to follow

Weaknesses:
W1) Experimental section does not fully convince of the validity of the method (see above)

**Questions For Authors:**

I would like the authors to comment on my questions about the experimental analysis

**Relation To Broader Scientific Literature:**

The theoretical contributions seem well contextualised in the literature. My only point is, as mentioned above, about additional experimental comparisons.

**Theoretical Claims:**

I checked the claims to the best of my abilities, however I could not verify the proofs due to my limited expertise in NTK theory.

---

> ### Author Rebuttal · Authors · 2025-04-01
>
> # Time and memory analysis
> ***
> Thanks for suggesting a more detailed time and memory requirements analysis. We conducted a measurement of the training time and memory of IGA across all experiments. For example, in the **1D simple function approximation**, the vanilla SIREN takes 28ms/iter (ms/iter denotes milliseconds per iteration) with 1009MB memory; SIREN+IGA takes 45ms/iter with 1575MB memory; adjustments (w/o IGA) by the full eNTK matrix, i.e., the eNTK matrix obtained from the population data, takes 198ms/iter with 5881MB memory. In the **2D image approximation**, the vanilla PE takes 74ms/iter with 3609MB memory; PE+IGA takes 87ms/iter with 4167MB memory. The full eNTK adjustment requires >100GB memory, which is practically infeasible. In the **3D shape representation**, the vanilla ReLU takes 113ms/iter with 2163MB memory; ReLU+IGA takes 163ms/iter with 2231MB memory. In the **5D neural radiance fields**, the vanilla NeRF takes 160ms/iter with 9451MB memory; NeRF+IGA takes 286ms/iter with 11033MB memory.
>
> The complete results can be found in: https://anonymous.4open.science/r/iga-inr-icml2025-3A4D/time_memory.md. Overall, IGA averages 1.56× the training time and 1.34× the memory of the baseline but achieves greater improvements (on average, **2.0× those of prior FR and BN**, as shown in the main text). Compared to the full eNTK matrix, IGA **saves at least 1/4 of the training time and memory**. Thus, our method improves fitting accuracy by enhancing training dynamics without significantly increasing time or memory. Notably, since IGA achieves the same results with fewer iterations, it often allows us to achieve results comparable to baseline methods in less or even shorter time. We present examples in the above link.
> # Practical issue
> ***
> Enhancing network performance by improving training dynamics without affecting the inference holds significant practical value. It helps improve inference performance in cases with limited computational resources. NTK theory has made significant progress in analyzing the training dynamics and spectral bias of INR and shows the theoretical potential to adjust dynamics. However, directly applying it to most INR models for better performance remains impractical due to the intractability of NTK matrix. **IGA provides two practical solutions**: first, it proves and validates the effectiveness of eNTK (which is generally computable) in adjusting training dynamics; second, it introduces inductive generalization of the transformation matrix derived from the eNTK matrix of sampled data to adjust the overall training dynamics. With these two solutions, IGA achieves greater improvements (on average, **2.0× those of prior FR and BN**, as shown in the main text) without significant additional costs—requiring only 45% more training time and memory on average compared to the baseline. Please note that FR and BN also cost more training time and memory but with only minor improvement. Thus, IGA is a practical and effective training dynamics adjustment method for INR tasks.
> # Additional experiments
> ***
> Thanks for suggesting more models to further show IGA's practical usability. As you mentioned, we have included 2D results of IGA with Gauss, WIRE, FINER in our Appendix and now provide 3D shape results. Due to time constraints and considering that MFN is the backbone of BACON, we extend IGA to MFN for both 2D and 3D tasks. Furthermore, for the neural radiance fields, we extend IGA to the well-known DVGO. The results are listed in the table of the following link: https://anonymous.4open.science/r/iga-inr-icml2025-3A4D/new_exp.md. Previous training dynamics methods, i.e., FR and BN, are also compared. From the numerical results, **IGA consistently outperforms FR and BN across 2D, 3D, and 5D tasks**, aligning with the main paper. These results further support our claims. We will add these results in the main paper.
> # Chamfer Distance
> ***
> Thanks for insightful comments about 3D shape representation. Following your suggestion, we computed Chamfer Distance using the code from “Occupancy Networks” and reported the results in the link: https://anonymous.4open.science/r/iga-inr-icml2025-3A4D/chamfer_distance.md. As you mentioned, Chamfer Distance and IoU can behave differently. Despite this difference, **IGA still outperforms the baseline and other training dynamics adjustment methods, FR and BN.**
> # Discussion on 3D Shape Representation
> ***
> Thanks for valuable suggestions on the 3D task. The setting of 3D shape representation in the main text is simple but has been adopted by many classic INR works to test the representation capability of INR models, such as PE, SIREN, MFN, Gauss, WIRE, and FINER. Recent works on training dynamics of INR, such as FR and BN, have also used this setting to evaluate the effectiveness of their methods. Therefore, this setting is suitable for demonstrating that IGA can better improve training dynamics to enhance INR’s representation performance for 3D shapes.

---

### Decision · Program_Chairs · 2025-05-01

**Decision:**

Accept (poster)

**Comment:**

This paper introduces a Neural Tangent Kernel (NTK)-based approach, the so-called Inductive Gradient Adjustment (IGA) method, to address the spectral bias in implicit neural representations (INRs).
The effectiveness of IGA is demonstrated across a wide range of INR tasks.

The authors have done an excellent job during the rebuttal phase, and all reviewers are now convinced of the significance and merit of the contribution, which clearly improves upon existing methods.

As a result, I recommend acceptance of this paper to ICML 2025.

A few minor suggestions for the final version:
- Please consider using proper LaTeX formatting for math operators: use $\sin$, $\min$, $\max$ instead of $sin$, $min$, $max$.
- Please include a time and memory analysis during rebuttal, along with implementation details, to complement the experimental results.
- I personally found it a bit difficult to clearly identify the proposed IGA method in Section 3.2, maybe consider adding an algorithm block or a summary table to make it easier for readers to follow.